

# DCG-MIP: The Debris-Covered Glacier melt Model Intercomparison exPeriment

Francesca Pellicciotti*[1,2#], Adrià Fontrodona-Bach*[1,2#], David R. Rounce[3], Catriona L. Fyffe[1,4#], Leif S. Anderson[5], Álvaro Ayala[1,6], Ben W. Brock[4], Pascal Buri[2,7], Stefan Fugger[2], Koji Fujita[8], Prateek Gantayat[1,9#], Alexander R. Groos[10,11], Walter Immerzeel[12], Marin Kneib[2,13], Christoph Mayer[14], Shelley MacDonell[6,15], Michael McCarthy[1,2], James McPhee[16,17], Evan Miles[2,18,19], Heather Purdie[20], Ekaterina Rets[21], Akiko Sakai[8], Thomas E. Shaw[1], Jakob Steiner[12,22,23], Patrick Wagnon[24], Alex Winter-Billington[25]

*These authors contributed equally to this work.
#Formerly at:
[1]Institute of Science and Technology Austria, Klosterneuburg, Austria;
[2]Swiss Federal Institute for Forest, Snow and Landscape Research WSL, Birmensdorf, Switzerland
[3]Department of Civil and Environmental Engineering, Carnegie Mellon University, Pittsburgh, PA, USA
[4]School of Geography and Natural Sciences, Northumbria University, Newcastle-Upon-Tyne, UK
[5]Department of Geology and Geophysics, University of Utah, Salt Lake City, UT, USA
[6]Centro de Estudios Avanzados en Zonas Áridas (CEAZA), La Serena, Chile
[7]Geophysical Institute, University of Alaska Fairbanks, Fairbanks, AK, USA
[8]Graduate School of Environmental Studies, Nagoya University, Nagoya, Japan
[9]Lancaster Environment Center, Lancaster University, Bailrigg, Lancaster, UK
[10]Institute of Geography, University of Bern, Bern, Switzerland
[11]Institute of Geography, Friedrich-Alexander-Universität Erlangen-Nürnberg, Erlangen, Germany
[12]Department of Physical Geography, Utrecht University, Utrecht, The Netherlands
[13]Laboratory of Hydraulics, Hydrology and Glaciology (VAW), ETH Zurich, Zurich, Switzerland
[14]Geodesy and Glaciology, Bavarian Academy of Sciences and Humanities, Munich, Germany
[15]Waterways Centre, University of Canterbury, Christchurch, New Zealand
[16]Department of Civil Engineering, University of Chile, Santiago, Chile
[17]Advanced Mining Technology Center, University of Chile, Santiago, Chile
[18]Glaciology and Geomorphodynamics Group, University of Zurich, Zurich, Switzerland
[19]Department of Geosciences, University of Fribourg, Fribourg, Switzerland
[20]School of Earth & Environment, University of Canterbury, Christchurch, New Zealand
[21]Institute of Geophysics, Polish Academy of Sciences, Warsaw, Poland
[22]Himalayan University Consortium, Lalitpur, Nepal
[23]Institute of Geography and Regional Science, University of Graz, Graz, Austria
[24]Univ. Grenoble Alpes, CNRS, IRD, IGE, Grenoble, France
[25]Te Puna Pātiotio Antarctic Research Centre, Te Herenga Waka Victoria University of Wellington, New Zealand

*Correspondence to*: Francesca Pellicciotti (francesca.pellicciotti@ista.ac.at) and Adrià Fontrodona-Bach (adria.fontrodona-bach@ista.ac.at)

**Abstract.** In a warming world of glacier changes, the scientific community has dedicated increasing attention to debris-covered glaciers and their response to climate. A variety of models with distinct complexity and data requirements have been developed and widely used to simulate melt under debris at different sites and scales, but their skills have never been compared. As part of the activities of the International Association of Cryospheric Science (IACS) Debris Covered Glacier Working Group, we present an intercomparison exercise aimed at advancing our understanding of model skills in simulating ice melt under a debris layer. We compare 14 models with different complexity at nine sites in the European Alps, Caucasus, Chilean Andes, Nepalese Himalaya and the Southern Alps of New Zealand, over one melt season. We run the models with measured meteorological data from automatic weather stations and estimated or measured debris properties. We consider four main model categories: i) energy balance models that calculate melt by solving the physics of heat transfer to the debris layer, but require a high amount of input data; ii) a simplified energy balance model; iii) an enhanced temperature-index model; and iv) simple empirical temperature-index models that have been extensively used given their low data requirement but require calibration of their



empirical parameters. Model performance is evaluated using on-site measurements of sub-debris melt (for all models) and surface temperature (for models based on the surface energy balance). Our results show that physically-based energy balance models and empirical temperature-index models perform in a distinct manner. At the one end of the spectrum, simple temperature index models are accurate when recalibrated or when using site-specific literature parameters, and show poor results when parameters are uncalibrated. At the other end, energy balance models show a range of performance: the most accurate energy balance models are those with the highest degree of complexity at the atmosphere-debris interface. An important data gap emerged from our experiment: the poor performance of all models at three sites was related to the poor knowledge of debris properties, and specifically of thermal conductivity. Future work should focus on both: i) consistent data acquisition to evaluate existing models and support new model developments; ii) advancing models by accounting for processes such as debris-snow interactions, moisture in the debris and refreezing. We suggest that a systematic effort of model development using a common model framework could be carried out in phase II of the Working Group.

## 1. Introduction

Glacier ice is often covered by a continuous or discontinuous layer of rock debris, which can vary in thickness from a few centimetres to several metres (Østrem, 1959; Kirkbride and Dugmore, 2003; Reid et al., 2012; Juen et al., 2014; Rounce and McKinney, 2014; Fyffe et al., 2020). Such debris-covered ice is extensive in many mountain ranges around the world (Scherler et al., 2018, Herreid and Pellicciotti, 2020). In a warming climate, debris cover has been observed to increase in area and thickness (Deline, 2005; Stokes et al., 2007; Bhambri et al., 2011; Thakuri et al., 2014; Mölg et al., 2019; Tielidze et al., 2020; Xie et al., 2020; Anderson et al., 2021) as a result of melt-out and accumulation of englacial debris at the glacier surface (Kirkbride and Deline, 2013; Anderson and Anderson, 2018) as well as increased debris input from surrounding slopes and lateral moraines destabilised by glacier debuttressing (van Woerkom et al. 2019) and permafrost degradation (Gruber et al., 2017). During sustained periods of negative glacier mass balance, debris cover expands laterally from medial moraines and upstream as debris-rich ice is brought to the surface (Anderson, 2000; Jouvet et al., 2011; Rowan et al., 2015).

The role of supraglacial debris in modulating glacier response to climate across scales is an open topic of research. We broadly understand debris to reduce melt rates when thicker than a few centimetres (Østrem, 1959; 1965; Kirkbride and Dugmore, 2003) and to potentially increase melt rates when thinner (Østrem, 1959; Mattson et al., 1993) or patchy (Fyffe et al., 2020). The relationship between debris thickness and sub-debris melt is commonly referred to as an Østrem curve, which has has been established through field observations (Østrem, 1959; 1965; Khan, 1989; Mattson et al., 1993; Konovalov, 2000; Popovnin and Rozova, 2002; Lukas et al., 2005; Mihalcea et al., 2006; Nicholson and Benn, 2006; Hagg et al., 2008) and numerical simulations with energy balance models at the point scale (Reid and Brock, 2010; Wang et al., 2011; Brook et al., 2013; Lejeune et al., 2013; Evatt et al., 2015). Studies that document melt across debris-covered glacier surfaces beyond the point scale are scarcer (Reid et al., 2012; Vincent et al., 2016; Anderson et al., 2021; Steiner et al., 2021) and our understanding of glacier-scale ablation patterns is more limited.

Research on debris-covered glaciers has seen an enormous growth in the last decade. Novel lines of research include the first global mapping efforts of debris areal extent (Scherler et al., 2018; Herreid and Pellicciotti, 2020); determining the thickness of debris covering glaciers at local and regional scales (Schauwecker et al., 2015; Groos et al., 2017; McCarthy et al., 2017, 2022; Nicholson et al., 2018; Rounce et al., 2018; Rounce et al., 2021); understanding how debris is transported through the ice and affects glacier flow and geometry (Rowan et al., 2015; Anderson and Anderson, 2016; Banerjee, 2017; Wirbel et al., 2018; Scherler and Egholm, 2020; Kirkbride et al., 2023; Margirier et al., 2025); identifying the distinct large scale thinning patterns of debris-covered glaciers as compared to debris-free glaciers (Kääb et al., 2012, Brun et al., 2019); advancing our understanding of debris-covered glacier meteorology (Brock et al., 2010; Shaw et al., 2016; Steiner and Pellicciotti, 2016; Yang et al., 2017; Steiner et al., 2018; Bonekamp et al., 2020; Nicholson and Stiperski, 2020), surface properties (Nicholson





and Benn, 2013; Rounce et al., 2015; Miles et al., 2017; Quincey et al., 2017) and hydrology (Fyffe et al., 2020; Miles et al., 2020); and insights into the processes controlling debris-covered glacier mass balance and the role that surface features such as ice cliffs and ponds play in amplifying mass balance locally and at the glacier scale (Sakai et al., 2000, 2002; Han et al., 2010; Immerzeel et al., 2014; Reid and Brock, 2014; Buri et al., 2016a,2016b,2018; Thompson et al., 2016; Salerno et al., 2017; Miles et al., 2016, 2018; Brun et al., 2016, 2018; Watson et al., 2018; Mölg et al., 2019; Anderson et al., 2021).

Some of these new lines of research have exploited satellite observations of increasing resolution (Brun et al., 2018) as well as surveys from Uncrewed Aerial Vehicles (Immerzeel et al., 2014; Kraaijenbrink et al., 2016, 2018; Fyffe et al., 2020; Westoby et al., 2020; Bisset et al., 2022; Messmer and Groos, 2024), which allow processes of glacier mass loss, debris evolution and dynamics to be understood at high resolution. Others have focused on model developments (e.g. Buri and Pellicciotti, 2018; Potter et al., 2020) and new theoretical advances (e.g. Nicholson and Stiperski, 2020). Despite these tremendous advances, some of the basic aspects of debris-covered glacier processes and modelling remain elusive (e.g. debris sourcing and evolution over scales, numerical reconstruction of debris thickness across spatial and temporal scales, the future trajectory of debris covered glaciers at local and global scales). Numerous models have emerged to represent some aspects of this complexity (e.g. Buri et al., 2016a,b; Rowan et al., 2015; Wirbel et al., 2018) but in the case of ablation of ice covered by debris, our understanding of key processes is still lacking.

Models developed to simulate the ablation of debris-covered ice can be broadly grouped into **physically-based energy balance models** and **empirical temperature-index models**, with a number of intermediate models between the two categories. **Energy balance models** estimate the energy fluxes at the interface between the debris and atmosphere, within the debris and at the interface between the debris and ice. As a result, they are able to explain the physical processes causing melt. These models have been primarily applied at the point scale using data from on-site automatic weather stations (Nicholson and Benn, 2006; Reid and Brock, 2010; Lejeune et al., 2013; Rounce et al., 2015; Giese et al., 2020), where they can be forced with meteorological data measured within the glacier boundary layer. Energy balance models have also been applied using off-glacier and re-analysis data products (e.g. Rounce et al., 2015). While energy balance models are physically realistic relative to temperature index models, they require more input meteorological data, as well as knowledge of debris properties and physical parameters used to calculate the main energy fluxes. These debris and atmospheric parameters (such as debris thermal conductivity, debris porosity, debris albedo, surface roughness length and heat transfer coefficients) are difficult to constrain spatially and temporally even for individual glaciers and short (sub-annual) periods, especially at remote sites outside of the European Alps, where most previous research has been carried out (e.g. Nicholson et al., 2006; Brock et al., 2010). Energy balance models are thus less commonly applied at the glacier scale (e.g. Fyffe et al., 2014; Reid et al., 2012; Groos et al., 2017; Shaw et al., 2016). It is also accepted by now that sophisticated models forced with low-quality input data will produce poor simulations (Machguth et al., 2008; Anslow et al., 2008; MacDougall and Flowers, 2011; Gabbi et al., 2015; Shaw et al., 2016).

On the other side of the spectrum are **temperature-index (or degree-day) models**. These models, initially developed for clean ice, assume a linear relationship between the melt rate and air temperature above a given temperature threshold, typically near 0°C, such that the melt can be estimated using a multiplicative factor called the degree-day factor (Hock, 2003). Because most debris-covered areas are mantled in relatively thick debris which reduces ablation, the most common approach to account for debris in these models has been to reduce the degree-day factor, thereby reducing the melt rates (e.g. Immerzeel et al., 2012, 2013; Shea et al., 2015). However, complexity can be added by including parameterizations for other factors such as the radiation component (Carenzo et al., 2016). Degree-day factors for different debris thicknesses have been calculated from sub-debris melt rates and air temperature measurements (e.g. Kayastha et al., 2000; Mihalcea et al., 2006; Hagg et al., 2008; Wei et al., 2010; Brook et al., 2013; Juen et al., 2014), but knowledge of their spatial variation remains a challenge that limits this approach. Constraining the variation of degree-day factors in space and in time is an area of active research, which has



generated a number of variants of this approach (Anderson and Anderson, 2016; Carenzo et al., 2016; Winter-Billington et al., 2020). Degree-day factors cannot be measured directly in the field, and rely on calibration with in-situ measurements, challenging their transferability to other sites. Despite this, temperature-index models have seen successful applications at the glacier and regional scale because they are simple, computationally efficient and require only air temperature (occasionally incoming shortwave radiation) as input and a low number of parameters (e.g., Kraaijenbrink et al., 2017). In most cases temperature-index models are applied at daily or coarser temporal resolution.

Different types of models respond to distinct needs, data availability and purposes. Often, numerical model development has balanced complexity with applicability. Energy balance models provide an accurate representation of the complex physical processes driving melt under debris at the expense of high data requirements, while temperature-index models provide a wider applicability at the expense of simplicity in process representation. Crucially, modelling skills, model structures and approaches have rarely been systematically evaluated across a range of sites for debris-covered glaciers.

Given the growing recognition that debris cover plays an important role in glacier mass balance and evolution, a Debris Covered Glaciers Working Group was established within the International Association of Cryospheric Sciences (IACS) to foster knowledge sharing and address key knowledge gaps within the community. In this context, we have designed and carried out a model experiment to compare several types of models with different degrees of complexity for modelling melt under debris. Our motivation is to understand whether models of differing complexity agree, under which conditions each of them performs well, and to identify areas where model development is needed. The comparison is carried out at the point scale of automatic weather stations installed on nine debris-covered glaciers using 14 models that cover a wide spectrum of model structure and complexity, and we compare them in a systematic manner at all sites. Our specific objectives are:

1. to assess model performance at different sites
2. to identify the strengths and limitations of the different model categories
3. to advance our understanding of the impact of model choice on the accuracy and uncertainty of simulated melt.
4. to attribute differences in model performance to model physics, assumptions and/or inaccurate data

We first present the design of the intercomparison experiment, describe the study sites and input data, and provide an overview of the models. We then present and discuss the results of the intercomparison, the implications for modelling and data collection experiments, and conclude with recommendations for future work.

## 2. Experimental setup

Our intercomparison was conducted as an open experiment. Nine data provider groups and 13 modelling groups responded to the call for participation. Modellers were provided with standardised meteorological data at hourly resolution, debris properties, and validation datasets that included sub-debris melt rate from ablation stakes and/or from ultrasonic depth gauge measurements, and debris surface temperature data (where available) for 9 study sites (Table 1). Some of the temperature index models used the sub-debris melt and/or surface temperature data for calibration, and the calibration strategy for individual models was left to the modellers (see Sect. 4.1 on model calibration).

At each site, we carried out two melt modelling experiments:

1. With measured debris thickness; and
2. With variable debris thickness (with values of 1, 2, 4, 6, 8, 10, 12, 15, 20, 30, 50 and 100 centimetres) to derive Østrem curves.



We quantified uncertainty for each experiment using Monte Carlo simulations where we varied the debris properties of the energy balance models and the parameters of the temperature-index models (see Sect. 4.2 on model uncertainty).

Each model was run for the period of time that data was available, which varied by site (Table 1). We restricted the model simulations to the period without continuous snow cover on the debris surface to avoid issues associated with the choices each modeller made associated with snow on top of the debris, which would affect the comparison of the sub-debris melt. Nonetheless, each modeller had to decide how to deal with short, occasional periods of snowfall (see model descriptions in the Supplement Sect. 2).

**3.    Data and study sites**

The experiments were performed at nine study sites, which included three in the European Alps, two in High Mountain Asia, two in the Chilean Andes, one in the Caucasus and one in the Southern Alps of New Zealand (Fig. 1; Table 1). Study sites were selected based on availability of a complete set of meteorological data, debris properties and validation data. The nine glacier sites span a large range of elevations and climates, as well as debris thickness and morphologies (Fig. 2). Nevertheless,

we recognise that our sites do not include some critical regions where debris is abundant: including Alaska, Greenland, Peru and the tropical Andes, Patagonia and the Western regions of High Mountain Asia such as the Karakoram (Fig. 1).

For each study site, the following data were provided:

- Automatic Weather Station (forcing) data: air temperature (°C), relative humidity (%), wind speed (m s$^{-1}$) and direction (°), air pressure (hPa), shortwave and longwave radiation (incoming and outgoing, W m$^{-2}$), precipitation
(mm hr$^{-1}$), snow depth (cm), height of meteorological sensors (m). These data were provided at hourly resolution. Wind direction was not used by any model.
- Debris thickness ($h_d$) measured at the site.
- Debris properties that were measured, derived, assumed or optimised in a previous modelling exercise: surface roughness length ($z_0$), thermal conductivity ($k_d$), porosity ($\varphi$) and  emissivity ($\varepsilon$) (Table S1).
- Validation data: surface height change measurements, from either ablation stakes or ultrasonic depth gauge readings, and debris surface temperature measurements derived from outgoing longwave radiation.
- Metadata of the site.

The Supplement provides additional data and information from the study sites: a photo from each of the nine sites (Fig. S1), the values of debris properties at each site and whether they were measured, optimised, estimated or assumed (Table S1), the
195 uncertainty of the validation data measurements (Table S2), and a summary of mean measured meteorological data at each site (Table S3).



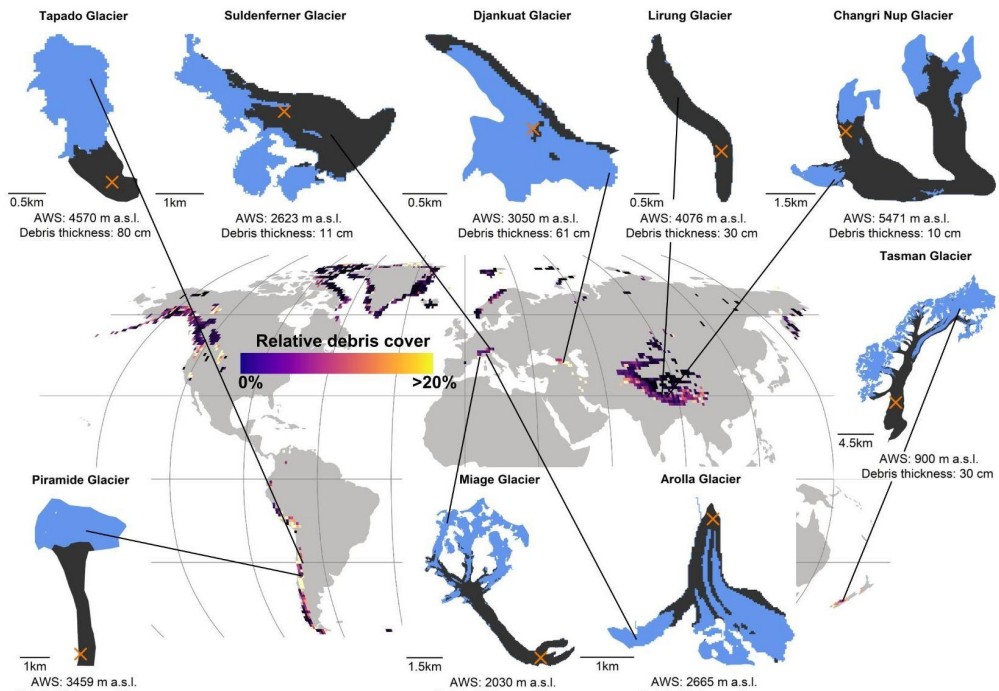

Figure 1. Location of the study sites and glacier maps with the position of the automatic weather station (indicated by a cross) on the debris-covered part (dark grey area) of each glacier (blue shade indicates clean ice). The background colours show relative debris cover per 1x1° tile, from Scherler et al. (2018).

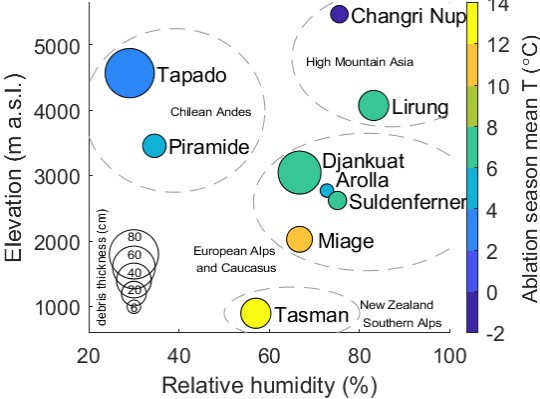

Figure 2. Characteristics of the study sites. Elevation, mean relative humidity, mean air temperature and debris thickness at the nine study sites, grouped by four main geographic regions. Circle sizes denote debris thickness at the location of the respective weather station. Mean air temperature and relative humidity are calculated as the average over the simulation period of the model intercomparison. Note that periods are of different duration at different sites (see Table 1).



**Table 1.** Overview of study sites. Validation data indicates what kind of melt observations are used at each site: ultrasonic depth gauge (UDG), ablation stakes, a draw-wire, and debris surface temperature ($T_s$). $h_d$ = debris thickness.

| Glacier | Latitude | Longitude | Elevation (m asl) | Start date | End date | Simulation length (days) | $h_d$ (cm) | Validation data |
|---|---|---|---|---|---|---|---|---|
| Arolla | 45.9814 | 7.5244 | 2665 | 04/07/2010 | 12/09/2010 | 71 | 6 | UDG, Stakes, $T_s$ |
| Changri Nup | 27.9925 | 86.7799 | 5471 | 16/04/2016 | 24/07/2016 | 100 | 10 | UDG, $T_s$ |
| Djankuat | 42.7551 | 43.2024 | 3050 | 17/06/2007 | 07/09/2007 | 83 | 61 | UDG, $T_s$ |
| Lirung | 28.2326 | 85.5621 | 4076 | 04/05/2014 | 24/10/2014 | 174 | 30 | UDG, Stakes, $T_s$ |
| Miage | 45.7823 | 6.8763 | 2030 | 13/06/2005 | 07/09/2005 | 87 | 22 | Stakes, $T_s$ |
| Piramide | -33.5896 | -69.8907 | 3459 | 21/10/2014 | 30/04/2015 | 192 | 18 | Stakes, $T_s$ |
| Suldenferner | 46.4958 | 10.5692 | 2623 | 19/07/2016 | 11/09/2016 | 55 | 11 | UDG, $T_s$ |
| Tapado | -30.1565 | -69.9224 | 4570 | 13/12/2014 | 22/03/2015 | 100 | 80 | UDG, $T_s$ |
| Tasman | -43.6286 | 170.2030 | 900 | 06/10/2017 | 09/04/2018 | 186 | 30 | Draw-wire, Stakes, $T_s$ |

## 4. Models

**Fourteen models** are part of our intercomparison experiment. When accounting for both calibrated and uncalibrated model runs, the total number of approaches increases to seventeen. Of the 14 models, nine were published prior to the call, three were a modification of a published model, and two were unpublished. The 14 models span a range of model complexity from energy balance models to temperature index models including intermediate models. The two intermediate models are different enough to be included in two distinct categories: a model that is close to the energy balance approach (a simplified energy balance) and a model that advances on the standard temperature-index approach (enhanced temperature index model). We thus group all models in **four categories**: eight energy balance models, one simplified energy balance model, one enhanced temperature index model and four temperature index models (Fig. 3). Temperature index model simulations were provided both in the uncalibrated and calibrated version where possible.

We rank models by complexity assuming that the temperature-index models are the simplest and the energy balance models the most sophisticated ones in the sense that they include the most complete representation of the physics of the processes leading to melt under debris. A general description of the model approaches can be found below, and an overview of participating models is shown in Table 2 and Fig. 3. A more detailed description of each model can be found in the Supplement, which fully documents unpublished approaches and includes sufficient detail of published models to enable understanding of the models' main characteristics and of the differences that are relevant for this intercomparison.

The models will hereafter be referred to by the model acronyms in Table 2. We used the model acronym provided in previous publications, or, when no name was provided, by the first three letters of the first author's last name followed by the publication year.



**Table 2.** Overview of the models used in this intercomparison, sorted alphabetically within each model category. Note models GRO17$_{A/B}$ are together as they contain the same information in this table. The Supplement describes further model details. EB = energy balance model; SEB = simplified energy balance model; ETI = enhanced temperature index model; TI = temperature index model. Ta = Air temperature, p = air pressure, RH = Relative humidity, FF = Wind speed, S↓, S↑ = Incoming/outgoing shortwave radiation, L↓ = Incoming longwave radiation, P = precipitation. NA = not applicable.

| Model name | Model type | Meteorological inputs | Temporal Resolution | Number of debris layers (N) Thickness of debris layer ($h_l$) | Model references |
|---|---|---|---|---|---|
| A-Melt | EB | T$_a$, RH, FF, S↓, S↑, L↓, P | hourly | N = 1 | Rets and Kireeva (2010), Elagina et al. (2025) |
| d2EB | EB | T$_a$, RH, FF, S↓, S↑, L↓, p | hourly | N layers of 1 cm each | Reid and Brock (2010), Steiner et al. (2018, 2021) |
| DEB$_{CF}$ | EB | T$_a$, RH, FF, S↓, S↑, L↓, P, p | hourly | N = $h_d$/10 | Reid and Brock (2010), Fyffe et al. (2014) |
| DEB$_{PG}$ | EB | T$_a$, RH, FF, S↓, S↑, L↓, p | hourly | N = $h_d$/3 if $h_d \leq 6$, $h_l$ = 2—2.05 if $h_d > 6$ | Reid and Brock (2010) |
| GRO17$_{A/B}$ | EB | T$_a$, RH, FF, S↓, S↑, L↓, p | hourly | N = 1 | Evatt et al. (2015), Groos et al. (2017a, b) |
| ROU15 | EB | T$_a$, RH, FF, S↓, S↑, L↓, P | hourly | N = $h_d$/10 | Rounce et al. (2015) |
| THRED | EB | T$_a$, RH, FF, S↓, S↑, L↓, P | daily | N = 1 | Fujita and Sakai (2014) |
| MCC19 | SEB | T$_a$, S↓, S↑ | hourly | N = $h_d$/10 if $h_d < 10$ cm, N layers of 1 cm if $h_d > 10$ cm | McCarthy (2025) |
| DETI$_m$ | ETI | T$_a$, S↓, S↑ | hourly | NA | Carenzo et al. (2016), modified |
| DDF$_{debris}$ | TI | T$_a$ | hourly | NA | Kayasha et al. (2000) |
| Hyper-fit | TI | T$_a$ | hourly | NA | Anderson and Anderson (2016), Anderson et al. (2021) |
| KO2 | TI | T$_a$ , S↓, S↑ | daily | NA | Winter-Billington et al. (2020) |
| KP1 / KM1 | TI | T$_a$ | daily | NA | Winter-Billington et al. (2020) |





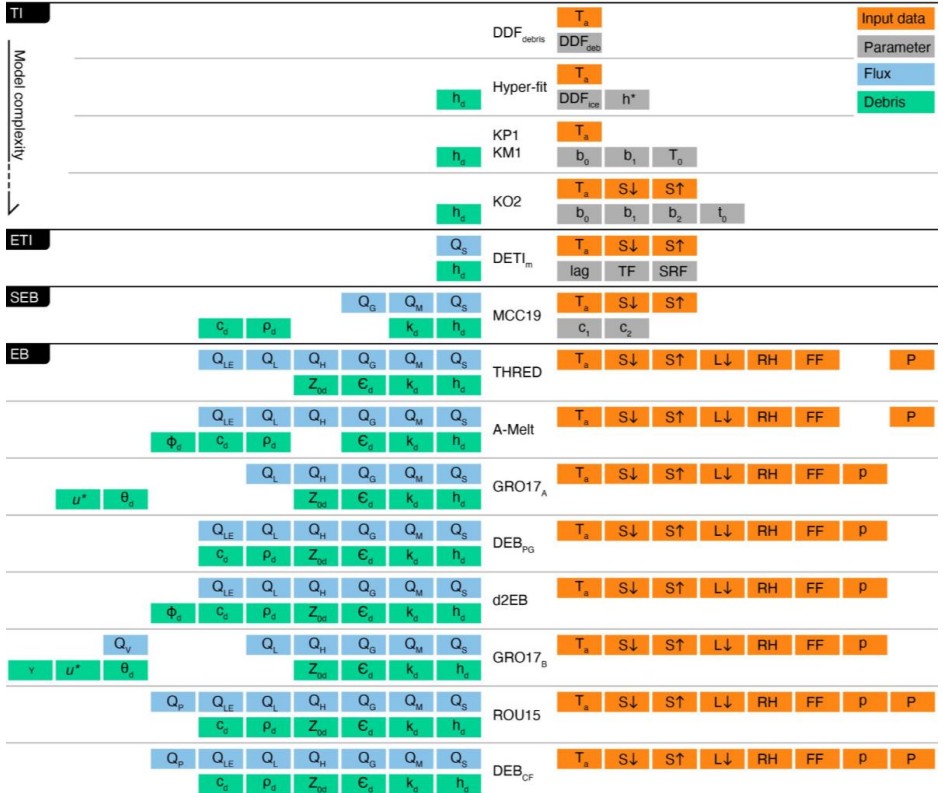

**Figure 3. A graphical representation of model complexity, from top (simplest models) to bottom (most sophisticated models), for all**
230 **four model categories considered in this intercomparison experiment (TI: temperature index models; ETI: enhanced temperature index model; SEB: simplified energy balance model; EB: energy balance models). Model complexity includes the required meteorological data (orange), empirical parameters (grey), debris properties (green) and energy fluxes (blue). Symbols for the meteorological input data are as in Table 2. Abbreviations and symbols for the fluxes, input data and parameters are as in the text. More details about models are in the Supplement.**

**Energy balance models**

*General model approach*

Energy balance models calculate sub-debris melt by solving two main equations: i) the heat exchange at the debris-atmosphere interface and ii) the heat conduction of this surface energy into the debris, until the energy reaches the debris-ice interface and is transferred to the ice. If the ice is at 0°C, this energy is used for melt; otherwise, energy is used to warm the ice towards 0°C.
240 This assumes that no other energy transfer occurs within the debris.

The general debris surface energy balance equation, following the notation of Reid and Brock (2010), is:

$$S{\downarrow} + S{\uparrow} + L{\downarrow} + L{\uparrow}(T_s) + H(T_s) + LE(T_s) + G(T_s) + P(T_s) = 0 \qquad (1)$$

where *S is the net shortwave radiation*, $L{\downarrow}$ *and* $L{\uparrow}$ are the incoming and emitted longwave radiation, *H* is the turbulent sensible heat flux, *LE* is the turbulent latent heat flux, *P* is the heat flux due to precipitation, *G* is the heat conducted into the debris
(equivalent to $G_1$, heat conducted into the first debris layer, in Fig. 4) and $T_s$ is the surface temperature. The fluxes that are a function of the surface temperature $T_s$ are explicitly indicated as such.



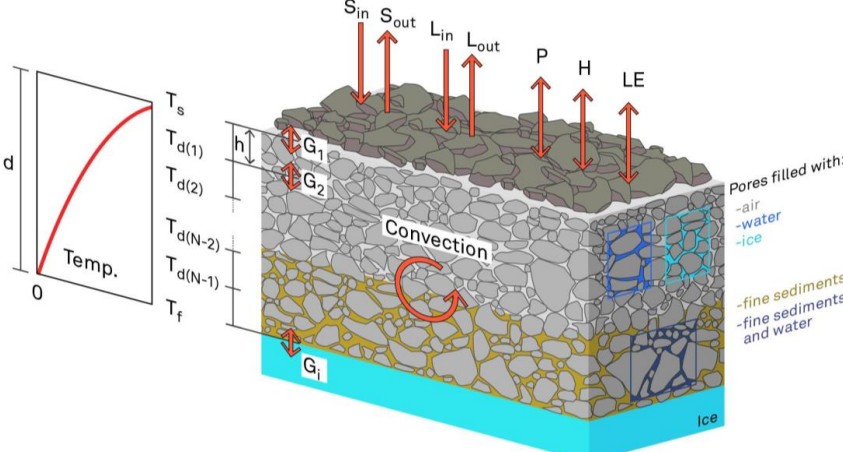

**Figure 4. Scheme of energy fluxes at the interface air-debris, within debris and between debris and ice. Fluxes are considered as positive when directed towards the surface and negative when away from the surface. The debris is discretised into N layers of height (h) each. Symbols for the fluxes are as in the text.**

The conductive heat flux G is the heat transferred through the debris layer to reach the ice and melt it, and it depends on the properties of the debris layer. This flux is calculated with the heat conduction equation through the debris (Eq. S7), and most models solve it by iteratively computing the debris surface temperature to close the energy balance, unless they assume a linear temperature profile or steady-state conditions, in which case the simpler linear Eq. S3 is used. When solving iteratively, most models, building on Reid and Brock (2010), use an iterative Newton–Raphson method to calculate surface temperature (Eq. S6), where the debris temperature is calculated for N layers of thickness $h$, with boundary conditions defined by the surface temperature, $Ts$, and the temperature of the debris/ice interface, which is assumed to stay at $T_f = 0°C$ for all models except one (A-melt).

The two main equations of the surface energy balance and heat conduction through the debris are similar for most models. Models however diverge in several aspects: i) the actual number of fluxes that are included; ii) the way individual fluxes are calculated, which can be more or less sophisticated; iii) the assumption about the temperature gradient in the debris; iv) the way the debris layer is discretised to calculate the conduction flux (i.e. the number of layers and their thickness); v) the numerical scheme to solve the two coupled equations above (see Supplement); vi) the ability to treat the interaction of debris with snow; and vii) the model temporal resolution (daily versus hourly).

*EB model complexity*

We use those aspects of model characteristics and representation of the debris domain to arrange energy balance models along an axis of complexity, from the simplest to the most sophisticated (Fig. 3). We also include in our arrangement the temporal resolution (hourly versus daily, with hourly models regarded as more complex). Since in this intercomparison ablation seasons with no snow cover (or only occasional snow cover) were chosen, the ability of the models to deal with snow is not taken into account. The overall definition of model complexity that includes also temperature-index and intermediate models is discussed at the end of this section (and also illustrated in Fig. 3).

All energy balance models calculate the net shortwave and longwave radiative fluxes and the turbulent sensible heat flux at the surface (Fig. 3). All models except one (GRO17) include the turbulent latent heat flux at the debris-surface, and two models include the heat flux due to rain (ROU15 and DEB$_{CF}$). Models differ substantially in how **turbulent heat fluxes** are calculated. Building on Kuzmin (1961) and Nicholson and Benn (2006), most models use simplified bulk approaches with constant



turbulent exchange coefficients (Steiner et al., 2018, 2021; Fujita and Sakai, 2014) and do not take into account atmospheric stability, thus assuming neutral conditions (Table S15 and S16). Only two models ($DEB_{CF}$ and $DEB_{PG}$) account for the stability of the atmosphere using non-dimensional stability functions for momentum and heat expressed as functions of the Richardson number (Reid and Brock, 2010; Fyffe et al., 2014). A second major difference is the way the relative humidity of the debris

surface is treated to calculate the latent heat flux. Since no data on the water content within the debris were available at any of the sites, modellers either neglected this flux or made assumptions on the actual relative humidity of the debris surface ($RH_s$) (Table S16). These vary from assuming that the surface is saturated when it rains ($DEB_{CF}$, ROU15) to assuming that $RH_s=100\%$ when the air relative humidity ($RH_a$) is 100% ($DEB_{PG}$).

Models also differ in how the debris layer is represented. Four models ($GRO17_A$, $GRO17_B$, THRED and A-Melt) assume the

debris can be treated as a single layer with a linear temperature gradient between the debris surface and the underlying ice (Table 2). This is an assumption that has been made for simulations with a time step of 24 hours, informed by temperature measurements showing that, although the profile is nonlinear at various times throughout the day, it is approximately linear on a 24 hour averaged basis (Nicholson and Benn, 2006; Brock et al., 2010; Reid and Brock., 2010). The only other study that used this assumption, Brock et al. (2010), assumes a linear temperature profile in simulations at 1 hour time-step, but introduces

a 'debris heat storage' flux to account for debris warming during the day and cooling at night. This assumption simplifies the calculation of heat conduction into the debris (Eq. S7), thereby considerably reducing the computational costs.

Four models ($DEB_{CF}$, ROU15, THRED, A-Melt) represent snow on the debris surface by calculating snowmelt until the debris is exposed again (Table S19). Even here, differences are evident: some models accumulate snow on the ground using the precipitation and air temperature record (THRED) while other models use the snow depth record to calculate snowmelt as long

as snow is on the ground ($DEB_{CF}$, ROU15).

There are other smaller differences between models. All models assume the ice to be at melting point, except for A-Melt (see Sect. 2 in the Supplement). All models use the thermal conductivity value provided for each site (Table S1) for the heat conduction flux calculations and the debris emissivity provided for the calculation of the outgoing longwave radiation flux. All models used the aerodynamic surface roughness length provided for the calculation of the turbulent heat fluxes, except the

A-Melt model which parameterised it based on Kumin (1961).

*Inclusion of convection within the debris*

All models, with the exception of $GRO17_B$, assume that energy conduction into the debris is the only mechanism by which heat is transported through the debris to the underlying ice. The turbulent latent heat flux within the debris - and its variation with depth - was introduced by Evatt et al. (2015) to reproduce the peak in melt rate associated with the critical debris thickness

(Østrem, 1959; Kirkbride and Dugmore, 2003; Reznichenko et al., 2010), and address a potential contradiction between model predictions and field observations: some field-derived Østrem curves show an initial increase in melt rates (compared to clean ice) up to a thickness of a few cms, after which melt decreases and reduces below the clean ice melt rate at a critical debris thickness (Østrem, 1991; Kirkbride and Dugmore, 2003). The only other work that attempted to reproduce this behaviour (Reid and Brock, 2010) attributed it to the patchiness of debris for a thin, non-uniform debris layer, and proposed a patchiness

parameter to mimic the increase in melt rate for thin debris. Evatt et al. (2015) instead included air flow through the porous debris layer, and accounted for the energy exchange between the moving air and the ice at the bottom of the debris layer that takes the form of either condensation or evaporation (turbulent latent heat flux). The airflow within the debris layer is attenuated with debris depth, causing a reduction in the evaporative heat flux as the debris thickens. This initially increases the melt rate, as less latent energy is used for evaporation and more energy becomes available for melting. However, as the debris

layer continues to thicken, its insulating effect eventually dominates, leading to a reduction in the melt rate. The $GRO17_B$



model builds on the Evatt et al. (2015) model to reproduce these processes within a porous debris layer. The model requires knowledge of porosity and grain size, which determine the friction velocity and wind speed attenuation parameters (Fig 3).

**Simplified energy balance model**

One simplified energy balance model (MCC19; McCarthy, 2025) is part of this intercomparison (Supplement Sect. 2.2). At the debris surface, the model computes the net shortwave radiative flux and the conductive heat flux, and represents the remaining fluxes using the air and debris temperature difference together with two free parameters (Fig. 3, Table S12) (cf. Oerlemans, 2001). The model conducts heat through the debris layer using the one-dimensional heat equation, where the boundary condition at the ice surface is the temperature of melting ice (following Reid and Brock, 2010), and the boundary condition at the debris surface is the simplified debris-surface energy balance. The model therefore only requires air temperature and incoming shortwave radiation as meteorological forcing and debris parameters (conductivity, heat capacity and density) to solve the heat conduction equation, and has two parameters that need to be calibrated. Melt is calculated from the conductive heat flux at the base of the debris layer.

**Enhanced temperature index model**

The debris enhanced temperature index (DETI; Carenzo et al., 2016) model was developed as a model of intermediate complexity between a temperature index model and an energy balance model, building on similar developments for clean ice (the ETI model, Pellicciotti et al., 2005). It includes the shortwave radiation balance, and a term dependent on air temperature that represents empirically all other fluxes in the energy balance equations. The model's empirical parameters are a function of debris thickness, to account for the time needed to transfer energy from the surface to the ice, and were derived through functional relationships between the shortwave radiation flux and temperature with sub-debris melt simulated by an energy balance model at different thicknesses. It is designed to run at hourly resolution.

**Temperature index models**

Temperature-index models assume that melt is linearly dependent on air temperature and use a degree-day factor to estimate melt. The degree-day factor is generally calibrated to reproduce observed melt under debris (e.g. Kayastha et al., 2000), and likely cannot be transferred to sites with a different debris thickness or different climates (Winter-Billington et al., 2020). Anderson and Anderson (2016) developed a sub-debris melt model (Hyper-fit; Anderson et al., 2021) where a degree-day factor for clean ice is used to estimate a hypothetical bare ice melt rate at each site. To estimate sub-debris melt, the bare ice melt rate is reduced based on local debris thickness and a characteristic debris thickness length scale, h*. The characteristic length scale controls how rapidly sub-debris melt asymptotes toward zero melt as debris thickens, via a hyperbolic relationship. The parameter h* can be calculated as a function of debris properties (conductivity and porosity, and ambient conditions) but the model performs best by constraining h* directly using empirical debris-thickness melt data. The model has two parameters: $DDF_{ice}$ and h* (Fig. 3, Supplement Sect. 2.4).

Winter-Billington et al. (2020) introduced two modifications of the temperature-index model by Kayastha et al. (2000), both designed for daily simulations. In the first one (KP1/KM1), the degree-day factor is computed as a function of debris thickness through an empirical relationship with seven parameters, while in the second model (KO2) the degree-day factor is a function of both debris thickness and potential incoming shortwave radiation that has four empirical parameters. The last model included in this intercomparison is the $DDF_{debris}$, the simplest degree-day factor approach calibrated for sub-debris melt reduction (Fig. 3).



**Sorting models by complexity**

Models have different levels of complexity based on the number of input data they require, the number of fluxes they calculate,
the physical realism of the equations used to calculate the fluxes, the assumptions made, the numerical schemes used, the
temporal resolution, the number of parameters and debris properties required and the vertical discretisation of the debris layer.
We have previously identified four model categories with varying levels of complexity. Sorting the complexity of models
**within each model category** is less straightforward. In order to use a definition that seeks to be the least subjective as possible,
we quantify the model complexity based on the sum of the total number of input data required, fluxes calculated, empirical
parameters and debris properties required. This is illustrated in Fig. 3. Based on this definition, the most complex models are
$DEB_{CF}$ and ROU15, with a total sum of 21 (8 input data, 7 fluxes, 6 debris properties), but we regard $DEB_{CF}$ as more complex
because it accounts for atmospheric stability corrections in the calculation of the turbulent fluxes. The simplest model is the
$DDF_{debris}$. We arrange the models along this axis of complexity in all figures throughout the manuscript.

### 4.1.    Model calibration

The energy balance models do not require calibration and were run with the input meteorological forcing and debris properties
provided per site. The only energy balance model that adopted a calibration strategy was $GRO17_B$ to account for the latent
heat flux within the debris and at the debris-ice interface in a realistic manner. Calculation of these fluxes requires porosity
and grain size, and lack of site-specific accurate field observations of these debris properties led to unrealistic turbulent heat
fluxes and poor simulation outcomes. Therefore, the turbulent heat fluxes from $GRO17_{A/B}$ were calibrated against the approach
of Nicholson and Benn (2006), to exclude combinations of model parameters (low friction velocity and rapid wind speed
attenuation) that lead to unrealistic turbulent heat fluxes (i.e. converging to zero). The simplified energy balance model,
enhanced temperature index model and other temperature index models were all calibrated.

For some of the latter models ($DETI_m$, KP1 and Hyper-fit), the **uncalibrated versions were also run** to assess their
transferability. The definition of uncalibrated was left to each modeller, so that in some cases literature parameters from the
sites were used as long as they were not optimised for the same time period. The choice of how to calibrate any  model was
considered part of the model setup, and was left to the individual modellers (parameters, goodness-of-fit metrics and target
variable). The entire period was used for calibration because the data series were not long enough to split into separate
calibration and validation subperiods. A brief overview is provided below and full details of the calibration procedures are in
the model descriptions in the Supplement.

The simplified energy balance model was calibrated using both the surface temperature and sub-debris melt data provided,
following a multiparameter, multiobjective optimisation approach (after Rye et al. 2010). The $DETI_m$ model was calibrated
against the $DEB_{CF}$ simulations, as in the original publication, while uncalibrated runs used the original model parameters from
Carenzo et al. (2016), which were obtained for Miage glacier. The calibrated version of Hyper-fit used hourly cumulative melt
data to optimise the characteristic length scale, h*, for each specific site and time period. The uncalibrated version of Hyper-
fit used previously-published, independent parameters from six of the nine sites (see Table S24). For the other three sites, the
global mean h* value (Anderson and Anderson, 2016) was used to represent h*. The KP1 model (in which the degree-day
factor depends on only the debris thickness) was run in its calibrated (KP1) and uncalibrated (KM1) version, with the calibrated
version optimising the threshold temperature $T_0$ using cumulative melt data while all other empirical coefficients are as in
Winter-Billington et al. (2020). The KO2 model (in which the degree-day factor depends on both debris thickness and net
shortwave radiation) is uncalibrated, as it uses the empirical coefficient from Winter-Billington et al. (2020).




### 4.2. Model uncertainty

Additional simulations were performed for the two experiments (see Sect. 2) and for all energy balance models to quantify uncertainty associated with debris properties. At each site, 100 samples were randomly taken from a uniformly distributed +/- 10% uncertainty range applied to surface roughness, debris thermal conductivity and debris porosity, while a +/- 5% range was applied to emissivity. These standardised uncertainty ranges match those provided for most properties at most sites, and are the same ranges used by Reid and Brock (2010). The parameters and corresponding uncertainty ranges are provided in Table S1. The standard deviation of the 100 simulations was used to estimate the uncertainty of modelled melt.

In an attempt to use an approach as similar as possible to the uncertainty in debris properties, all temperature index models used +/-10% range around the calibration range of model parameters at each of the nine sites, and 100 randomly sampled combinations of parameters within that range. Uncalibrated models used a +/-10% of the applied parameter values.

### 4.3. Model Evaluation

All models are evaluated against measured sub-debris ice melt, while only the energy balance models and the simplified energy balance model are assessed against the debris surface temperature too, as the other models do not calculate surface temperature.

#### 4.3.1. Evaluation against melt observations

Models were evaluated against observed sub-debris melt using daily surface height change measurements for Arolla, Changri Nup, Djankuat, Lirung, Suldenferner, Tapado and Tasman, and three discrete measurements of ablation stakes over the ablation season for Miage and Pirámide. The percentage final melt error, simply defined as the difference between modelled and observed melt at the end of the simulation period as a percentage of the observed melt, was used to evaluate the modelled melt. This metric allows for comparison across all sites regardless of the resolution and type of validation data available. We evaluate performance across models and sites based on the median percentage error and the interquartile range (spread). We follow a similar method to Farinotti et al. (2017) and rank the models' performance based on their ranking for median and interquartile range errors.

#### 4.3.2. Evaluation against debris surface temperature

Hourly and daily debris surface temperature was used to evaluate the models using the root mean square error (RMSE) and bias between modelled and observed surface temperature. The Nash-Sutcliffe Efficiency was also used to further evaluate the hourly performance of models because this metric is most sensitive and meaningful for clearly defined daily cycles. The observed debris surface temperature was derived at all sites from the measured outgoing and incoming longwave radiation (which were available at all sites), following Stefan Boltzmann law as:

$$T_s = \left( \frac{L\uparrow - ((1-\varepsilon) \cdot L\downarrow)}{\varepsilon \cdot \sigma} \right)^{1/4} \tag{2}$$

where $T_s$ is the surface temperature of the debris (in K), $\varepsilon$ is the emissivity of the debris, $\sigma$ is the Stefan-Boltzmann constant (in W m$^{-2}$ K$^{-4}$), and $L\uparrow$ and $L\downarrow$ the outgoing and incoming longwave radiation (in W m$^{-2}$), respectively.





## 5. Results

### 5.1. Performance of model ensemble at sites

Fig. 5a shows the ensemble mean daily melt compared to the observations at each site for all models considered. As expected, sites with thinner debris show higher daily melt rates than sites with thicker debris, with the exception of Changri Nup due to its low ablation season temperature (Fig. 2). The observed melt is within the ensemble range of modelled melt at all sites, but variations among sites and groups of models are strong. Relative errors and the spread of model performance become larger with increasing debris thickness, with the exception of Tapado. The ensemble performance of models can also be observed in Fig. S2, where the continuous cumulative melt for the entire period is shown for each site and model.

Three groups of sites are evident. First are **the alpine sites of Arolla and Suldenferner,** where models' performance is high and consistent for most energy balance and temperature-index models except for some uncalibrated ones. The models' median error is lowest at Suldenferner (-1.1%), followed by Arolla (-3.7%) (Table 3). For these sites, models are consistent and show the smallest spread in terms of interquartile range (14.7% for Suldenferner and 15.2% for Arolla, Table 3).

Second, at Changri Nup, Miage, Pirámide and Tapado (three of the four highest sites in elevation), models' performance is relatively high, with -7.5% median error for Tapado, 7.2% for Pirámide and a higher -15.8% at Changri Nup and 23.7% at Miage. These sites show a larger spread in model performance, with 24.2% at Changri Nup, 34.5% at Miage, 37.8% at Pirámide and 76.7% at Tapado, although the latter is because of the low absolute melt at this site, which makes the relative errors larger.

Finally, **three sites stand out as characterised by low performance**, high errors for most models and large spread among models: **Lirung, Tasman and Djankuat** (with median melt errors of 55.2%, 57.9% and 90.8% respectively, Table 3). However, the consistency of the large median errors across model groups differs at these three sites. At Tasman, the energy balance models show high consistency and are grouped together but highly overestimate the observed melt, while the temperature-index models perform better but have a much larger spread. On Lirung, the energy balance models also perform poorly and consistently overestimate melt, while the temperature-index models consistently show a smaller error but large spread. Finally, on Djankuat, all four groups of models perform poorly and in a comparable manner, with the energy-balance models having most of the largest errors and the temperature-index models the lowest errors. In general, these poor-performance sites have both high debris thickness (30, 30 and 61 cm for Lirung, Tasman and Djankuat, respectively, Fig. 5 and Table 1), and very high debris thermal conductivity (Table S1).



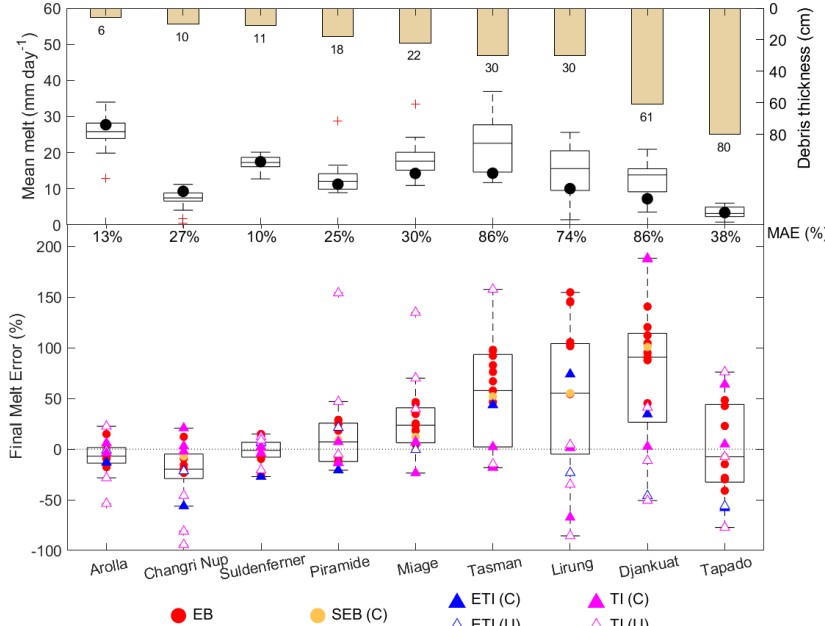

**Figure 5. Performance of the ensemble of models. (a) Modelled mean daily melt by the model ensemble (boxplot) and observed mean daily melt (black dot). Red crosses indicate outliers, defined as more than 1.5 times outside of the interquartile range. The model ensemble mean absolute error (MAE) is expressed as a percentage and shown for each site. Bar plots at the top indicate the debris thickness at each site. (b) Final melt error (in %, as defined in Section 4.3.1) at each site, for each group of models, and for calibrated and uncalibrated models separately. Sites are ordered by debris thickness.**

Validation against hourly surface temperature shows both similarities and differences from the validation against melt observations (Table S27, Fig. S3 and Fig. S4). Arolla, an Alpine site with good performance against melt observations, has the best performance (smallest RMSE) against debris surface temperature, with a median RMSE of 3.1°C. One of the worst performing sites against melt observations, Lirung, also has the worst performance against surface temperature, with a RMSE of 7.1°C, and the lowest Nash-Sutcliffe Efficiency (Fig. S4). Tasman is a distinct site: it shows a high performance against surface temperature with a RMSE of 3.8°C (Table S27), and a high and consistent NSE across models (Fig. S4), in contrast to the poor agreement with observed melt (Fig. 5 and Table 3). Contrasting to the high melt performance, most simulations in Tapado show a high RMSE, but also a high NSE, indicating the daily cycle of temperature is well reproduced despite a temperature bias (Fig. S3 and S4). The rest of the sites (Changri Nup, Djankuat, Miage, Piramide and Suldenferner) have a similar RMSE between 4.2 and 4.9°C (Table S27), despite their different performances against melt. Model consistency for surface temperature at sites is variable, very high in Arolla and Tasman, and low on Lirung and Tapado (Fig. S3).



**Table 3.** Modelled melt error across models and across sites in percentage. Models are ordered by complexity as defined in Figure 3. The last two columns correspond to the median and interquartile range (IQR) across sites per model, and the last two rows correspond to the median and IQR across models per site. We provide the IQR as a measure of the spread in model performance. "C" indicates calibrated, and "U" indicates uncalibrated.

| Category | Model | ARO | CNU | DJA | LIR | MIA | PIR | SDF | TAP | TAS | Median (%) | IQR (%) |
|---|---|---|---|---|---|---|---|---|---|---|---|---|
| EB | DEB$_{CF}$ | -17.7 | -23.3 | 45.4 | 103.7 | 4.6 | -7.2 | -25.7 | -41.0 | 46.5 | -7.2 | 69.6 |
| EB | ROU15 | -9.6 | -22.4 | 87.8 | 103.1 | 18.4 | -7.5 | -5.8 | -7.7 | 67.1 | -5.8 | 80.4 |
| EB | GRO17$_B$ | -6.9 | -23.4 | 120.6 | 154.8 | 46.5 | 29.0 | 15.0 | 48.5 | 98.1 | 46.5 | 94.2 |
| EB | d2EB | -16.0 | -15.8 | 90.8 | 105.7 | 24.2 | -11.7 | -5.3 | -28.3 | 57.9 | -5.3 | 82.0 |
| EB | DEB$_{PG}$ | 14.9 | -19.7 | 95.2 | 54.4 | 23.7 | 18.3 | -9.5 | -29.8 | 76.3 | 18.3 | 72.0 |
| EB | GRO17$_A$ | 0.2 | 12.2 | 112.4 | 145.1 | 44.8 | 28.5 | 13.9 | 42.7 | 97.3 | 42.7 | 87.6 |
| EB | A-Melt | -9.2 | -9.2 | 104.6 | 101.8 | 25.4 | -13.2 | -7.3 | -14.8 | 92.2 | -7.3 | 104.8 |
| EB | THRED | -7.1 | -5.7 | 140.8 | 146.1 | 34.7 | 24.8 | 4.0 | 22.9 | 82.9 | 24.8 | 95.8 |
| SEB | MCC19-C | -1.6 | -7.1 | 100.9 | 55.2 | 12.1 | 9.3 | -1.1 | -7.4 | 52.2 | 9.3 | 55.9 |
| ETI | DETIm-C | -13.0 | -56.2 | 34.6 | 74.0 | -0.5 | -20.7 | -27.1 | -58.0 | 43.6 | -13.0 | 71.2 |
| ETI | DETIm-U | -1.0 | -21.5 | -46.0 | -23.3 | -0.5 | 21.4 | 6.2 | -56.1 | -18.0 | -18.0 | 30.2 |
| TI | KO2-U | -28.3 | -81.1 | -50.7 | -85.5 | 70.2 | 46.9 | 12.2 | -77.2 | 157.8 | -28.3 | 130.9 |
| TI | KM1-C | -3.7 | -2.1 | 2.6 | -67.3 | -23.5 | 7.2 | -4.0 | 4.9 | -18.2 | -3.7 | 22.7 |
| TI | KP1-U | -53.6 | -94.3 | -11.3 | -34.8 | 134.9 | 154.2 | -20.5 | -7.5 | 542.6 | -11.3 | 179.2 |
| TI | Hyper-fit-C | 6.1 | 3.1 | 188.5 | 1.2 | 7.3 | -13.6 | 2.5 | 63.9 | 2.1 | 3.1 | 19.5 |
| TI | Hyper-fit-U | 22.6 | -45.8 | 41.3 | 4.3 | 39.5 | -5.0 | 8.8 | 76.2 | -14.7 | 8.8 | 47.4 |
| TI | DDF$_{debris}$-C | 6.0 | 20.8 | 187.6 | 1.7 | 6.9 | -13.3 | 1.9 | 64.1 | 2.3 | 6.0 | 29.8 |
| - | Median (%) | -3.7 | -15.8 | 90.8 | 55.2 | 23.7 | 7.2 | -1.1 | -7.5 | 57.9 | 4.3 | 59.7 |
| - | IQR (%) | 15.4 | 24.2 | 87.8 | 109.1 | 34.5 | 37.8 | 14.7 | 76.7 | 91.2 | | |

Model performance against *daily* surface temperature is higher but results from averaging out of errors in the hourly time series (Table S27, Fig. S3). At this time scale, the overall pattern of performance across sites does not change substantially, but tends to be more uniform, suggesting that aggregation to daily resolution, by smoothing out variability and errors, is not appropriate to identify models' skills.

### Consistency of model ensemble performance

A key aspect of energy balance model performance is the accuracy in simulating both sub-debris melt and debris surface temperature. To evaluate this, we plot the melt error against the daily bias in surface temperature for each site separately (Fig. 6). A high model performance across both validation variables is indicated by models centred around the origin, and a low scatter indicates high consistency across models.

At most sites, the majority of models diverge from the origin (Fig. 6). A consistent warm bias in surface temperature is evident at most sites, with the exceptions of Piramide and Tasman, which exhibit a cold bias across models (Fig. 6). Models perform consistently well at Arolla and Suldenferner (low scatter) for both melt and surface temperature (Fig. 6e,g). At Piramide and Tapado, models are distributed across three different quadrants, indicating a variability in melt errors despite consistent surface temperature biases (Fig. 6f,h). Conversely, at Lirung and Djankuat, models consistently fall in the same quadrant, albeit with high scatter and high errors for melt at both sites and for surface temperature at Lirung (Fig. 6 c,d). At Tasman, models show a rather high consistency, with a high overestimation of melt and a cold temperature bias for most models (quadrant IV, Fig. 6i). The observed consistency in surface temperature biases can also be observed in Fig. S5, where all but two models show a positive temperature bias at most sites. The clustering in Fig. 6 also contrasts with the scatter in Fig. S5, and provides a clear



demonstration that the sites' characteristics or data quality likely control the spread of performance more than the model physics

(see discussion).

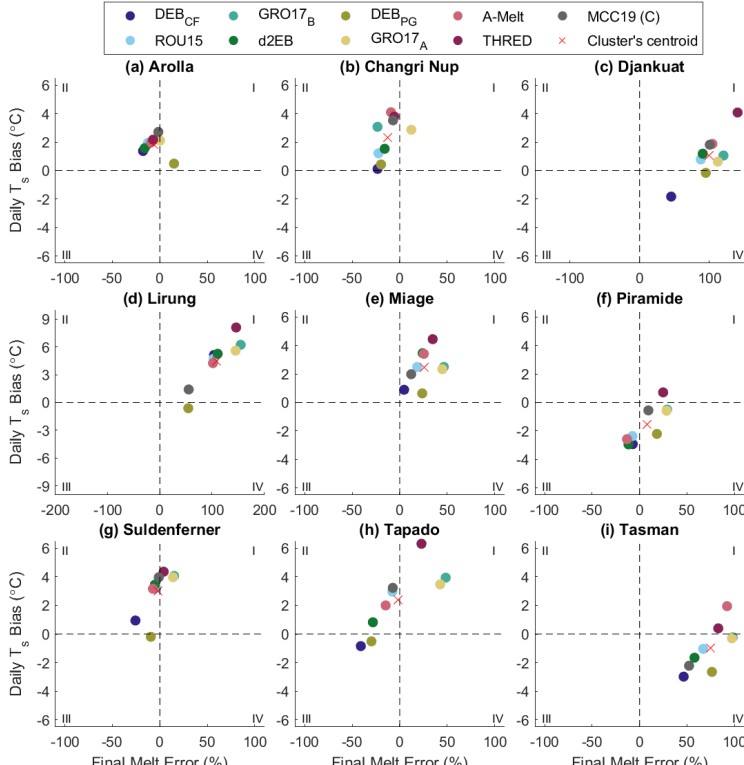

**Figure 6. Consistency of model performance across the two validation datasets. For each site (a-i), each energy balance model (and simplified energy balance) is scattered based on their daily temperature bias and melt error. Dashed lines correspond to the zero line for both axes, and separate the plot in four quadrants. Quadrants above the horizontal dashed line indicate overestimation of surface temperature. Quadrants to the right of the vertical dashed line indicate overestimation of melt. Note Djankuat and Lirung have wider axes ranges due to the high errors at those sites.**



## 5.2. Performance of individual models

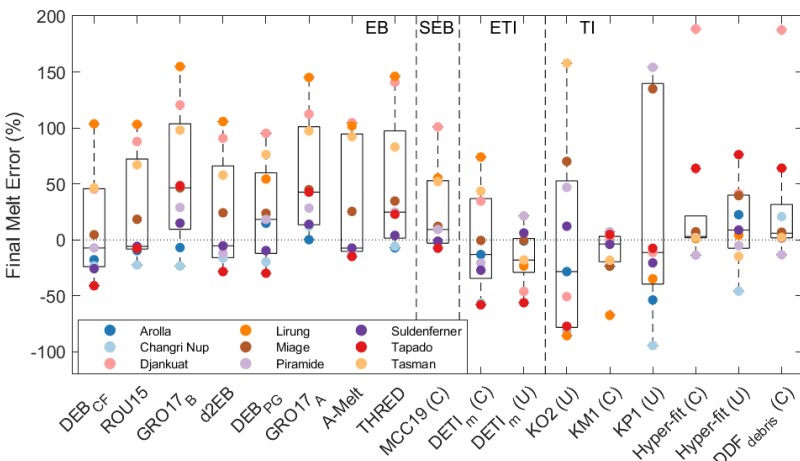

**Figure 7.** Final melt error (as defined in Section 4.3.1) for individual models at all sites expressed as interquartile box plots. Models are grouped in the four categories that we use in the paper (separated by dashed lines) and sorted from more complex (left) to less complex (right). Both calibrated (C) and uncalibrated (U) temperature index models are shown.

We evaluate the performance of individual models across sites against melt for all models (using the final melt error, Fig. 7, Tables 3 and 4), and against debris surface temperature for models solving the energy balance (using the root-mean-square-error, Fig. 8, Fig. S6). Overall, models tend to overestimate melt, with a median final melt error of 4.3% across all model runs (Table 3), dominated by the three sites where melt is consistently overestimated. However, there are important differences between models.

The highest ranked model (the calibrated Hyper-fit) and the lowest ranked model in Table 4 (uncalibrated KO2) are temperature-index models, highlighting that they perform very well when calibrated and show a much wider spread when uncalibrated. The calibrated temperature index models (calibrated Hyper-fit, DDF$_{debris}$ and KM1) have a low error and high consistency, here regarded as the spread (interquartile range) among sites. They are the first three models ranked on Table 4. The uncalibrated temperature index models have higher errors and lower consistency. The Hyper-fit model performs well when uncalibrated (Table 3) because it uses literature parameters for the specific study sites (even if not tuned for this experiment). KM1, KP1 and KO2 tend to underestimate melt, while the DDF$_{debris}$ and the Hyper-fit models tend to slightly overestimate it. These differences may result from differing parameter calibration choices. The DETI$_m$ model performance is unusual, as the uncalibrated version performs almost as well as the calibrated version on average, but has a smaller spread across sites. The calibrated model strongly overestimates melt at Lirung, Tasman and Djankuat, as it follows the performance of its reference EB model (DEB$_{CF}$ model, Sect. 4.2 and Supplement), and its performance reflects the performance of the DEB$_{CF}$ model and thus ranks in the middle on Table 4.

Clear differences are also evident among energy balance models. The models that perform best in terms of the smallest median error are DEB$_{CF}$ (-7.2% error, Table 3), ROU15 (-5.8%, Table 3) and d2EB (-5.3%, Table 3). The ones with the strongest consistency (taken as the IQR) across sites are DEB$_{CF}$ (69.6%, Table 3) and MCC19 (55.9%, Table 3), which is a calibrated model. This makes DEB$_{CF}$ the highest ranked energy balance model in Table 4, followed closely by ROU15 and d2EB. The energy balance model with the lowest performance in terms of median error is GRO17$_B$ (46.5% error, Table 3), which is also the lowest ranked energy balance model in Table 4, followed closely by THRED and GRO17$_A$.





**Table 4.** Ranking of model performance based on median and interquartile ranges. Models are given a rank for their median and their IQR performance on Table 3. Models are then sorted from highest to lowest overall (mean) rank.

| Model type | Model name | Rank Median | Rank IQR | Overall Rank |
|---|---|---|---|---|
| TI | Hyper-fit (C) | 1 | 1 | 1 |
| TI | KM1 (C) | 2 | 2 | 2 |
| TI | $DDF_{debris}$ (C) | 5 | 3 | 4 |
| EB | $DEB_{CF}$ | 6 | 7 | 6.5 |
| TI | Hyper-fit (U) | 8 | 5 | 6.5 |
| EB | ROU15 | 4 | 10 | 7 |
| EB | d2EB | 3 | 11 | 7 |
| SEB | MCC19 (C) | 9 | 6 | 7.5 |
| ETI | $DETI_m$ (U) | 12 | 4 | 8 |
| ETI | $DETI_m$ (C) | 11 | 8 | 9.5 |
| EB | A-Melt | 7 | 15 | 11 |
| EB | $DEB_{PG}$ | 13 | 9 | 11 |
| TI | KP1 (U) | 10 | 17 | 13.5 |
| EB | $GRO17_A$ | 16 | 12 | 14 |
| EB | THRED | 14 | 14 | 14 |
| EB | $GRO17_B$ | 17 | 13 | 15 |
| TI | KO2 (U) | 15 | 16 | 15.5 |


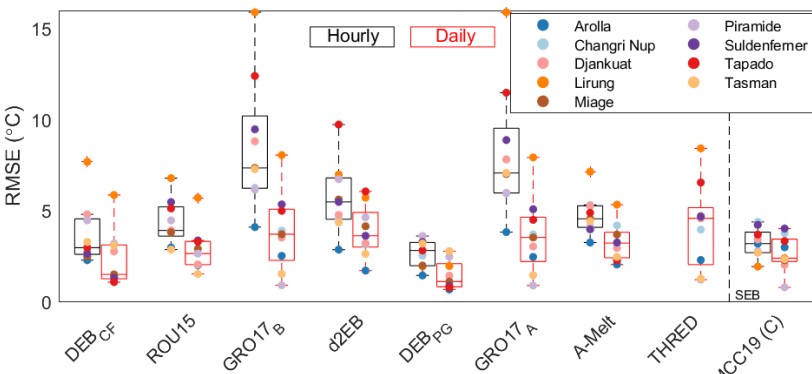

**Figure 8. Validation of the energy balance model simulations (including SEB) against surface temperature, at all sites. Statistical boxes for each model show the distribution of the Root Mean Square Error of surface temperature at each site. Black boxes are for hourly data, red boxes are for daily data. Note that the THRED model is run and validated at daily resolution only.**

The energy balance models (including the simplified energy balance model) were also evaluated against the surface temperature data (Fig. 8, Table S27). The RMSE is provided both at the hourly scale (of most simulations) and at the daily scale to include the THRED model. Important differences between energy balance models are apparent. At the **hourly scale**, $DEB_{CF}$ and $DEB_{PG}$ perform particularly well, with low median RMSE, of 3.0°C and 2.8°C (Fig. 8, Table S27), respectively, as well as high NSE of 0.81 and 0.88 respectively (Fig. S6), and a low spread and thus high consistency across sites. ROU15 and

A-Melt have a slightly higher median RMSE of 3.9°C and 4.6°C, respectively. The $GRO17_A$, $GRO17_B$ and d2EB models show the highest RMSE, of 7.2°C, 7.1°C and 5.5°C, and lowest NSE of 0.15, 0.27 and 0.52, respectively (Fig. S6), and a much larger spread across sites.

At the **daily scale**, the patterns of errors remain the same, but with overall lower RMSEs (Fig. 8, Table S27). The THRED model has the highest median RMSE (4.6°C). The simplified energy balance model performs well at both temporal scales



(RMSE of 3.2°C and 2.4°C, respectively), but this is to be expected given that it is calibrated against surface temperature. Sites with poor performance for more than one model are Tapado, Lirung and Piramide.

### 5.3.    Model uncertainty

We depict the uncertainty in model results due to debris properties and model parameters (see Sect. 4.3) in Fig. 9. Model sensitivity to parameter uncertainty is small (around 5% for most models) compared to the modelled total error for most models
considered and at most sites; particularly so for the energy balance models. For the energy balance models, the model sensitivity to uncertainty in debris properties propagates into final modelled melt uncertainty in a manner that is consistent both among models and sites. For empirical models, the comparison is more difficult, as each model has different parameters that control the melt calculations in a distinct manner; parameters have different meanings and units, and the plausible parameter ranges vary between parameters and are more difficult to define. Even with these caveats, however, model sensitivity
due to estimated parameter uncertainty is smaller than the model error for most temperature index models. In general, empirical models have higher uncertainty, and the uncertainty is more variable among models, reflecting models' differences from one another. The uncertainty is slightly higher for the KM1/KP1 models, followed by the $DETI_m$ model, while the lowest uncertainty is for the $DDF_{debris}$, followed by the Hyper-fit models.

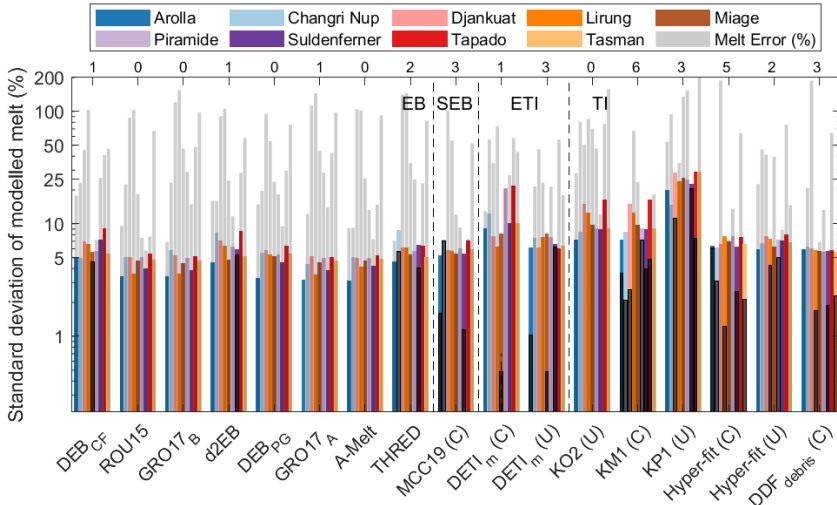

**Figure 9. Model sensitivity to parameter uncertainty (standard deviation of the Monte Carlo 100 model runs in %) for all models. The model error is depicted on the same scale as grey bars (total melt error in %, as shown in Fig. 5 and 7), for each model and site. Note the logarithmic y-axis. Where the model error is less than the uncertainty value, the grey bar denoting melt error overlaps the uncertainty bar and is therefore displayed darker. The number above each model bar denotes the number of sites (out of nine sites) in which the melt error is lower than the model uncertainty.**

### 5.4.    Østrem curves

All models until now have been evaluated based on their performance at the location of the automatic weather stations for the measured debris thickness. Here, we plot simulated melt as a function of debris thickness (as described in Sect. 2) (Fig. 10) and consider how divergent the models are when used to calculate melt at different thicknesses, which has implications for simulations of melt at the scale of an entire glacier, i.e. across a spatial domain of varying thickness.





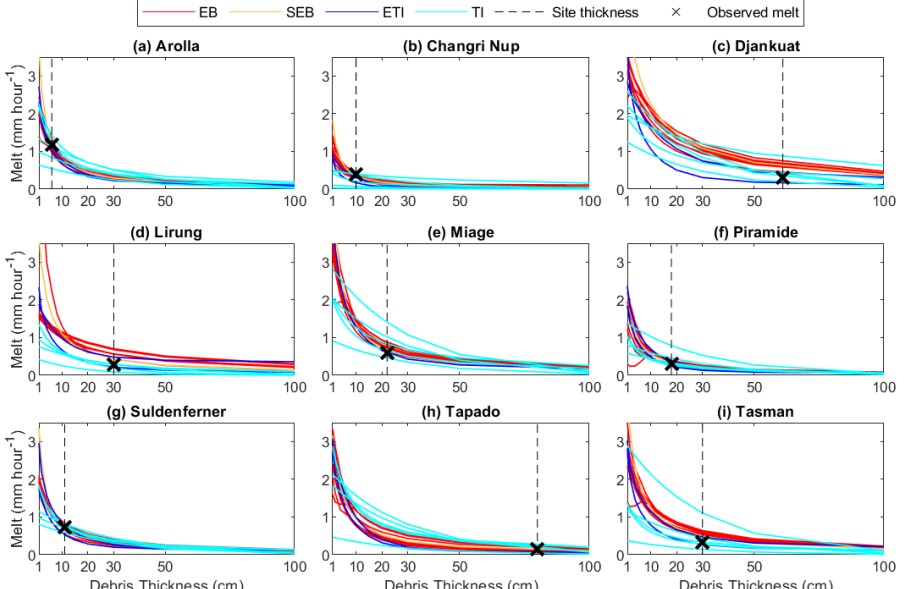

**Figure 10. Østrem curves for all models, aggregated into model groups. The actual thickness at each site is indicated by a vertical line, and the observed melt by a cross.**

Fig. 10 shows the Østrem curves simulated by each model, grouped by model type, and it highlights clear differences between models and sites (see also Fig. S7). The temperature-index models produce a larger spread for the same value of debris thickness than the other models and generally exhibit a more linear behaviour than the rest of the ensemble (Fig. S7), suggesting smaller sensitivity to debris thickness. This is not the case for the ETI and SEB, which simulate shapes that are close to those of the energy balance models. For most sites, and especially at Lirung, the spread among models increases dramatically for thin debris, implying that **the choice of model is crucial for melt simulations for thin debris**. It also suggests that we are not able to constrain the Østrem curves for thin debris, partly because this experiment is conducted with the input data and debris properties of the original location with thicker debris, which do not represent the properties of a very thin debris layer. Surface albedo in particular remained constant with debris thickness but should increase for thin debris as this becomes patchy and the debris cover area decreases (Azzoni et al., 2016). Models also diverge for thicker debris at Djankuat (Fig. 10b and c).

For most models and sites, the slope of the curve is steeper for thin debris than for thick debris (Fig. 10, S7), suggesting that models are **more sensitive where the debris is thinner**. This points to the importance of correctly representing processes where debris is thin (Fyffe et al., 2020), where we expect a quicker response to changing meteorological conditions, quicker drying or moistening of debris, and a larger role of surface roughness. Only the GRO17$_B$ model is able to reproduce the peak in melt occurring for thin debris based on the data provided and the experimental set up (Fig. S7). This melt enhancement is mostly visible at Tasman and Piramide and occurs only for thin debris.



### 6. Discussion

**6.1. Performance across sites and importance of debris properties and input data quality**

*6.1.1. Sites with high performance*

A clear result of our analysis is that model performance varies considerably between sites. Models perform **well and in a consistent manner at the three European Alps sites**: at Arolla and Suldenferner, with a consistent, high performance, and at Miage, with slightly larger errors (Sect. 5.3). For energy balance models, this might be due to a combination of three aspects: i) most energy balance models have been developed for initial application to those sites, and thus might be better suited to represent processes that dominate there; ii) the ease of access to these sites facilitates field visits, instrument maintenance and data quality checks, so that the quality of input and validation data might be higher; and iii) debris properties are better constrained at those sites as they have been measured there (e.g. Brock et al., 2010; Reid and Brock., 2010). At Miage, in particular, an extensive effort of field measurements since 2005 (Brock et al., 2010) has made this glacier one of the few where debris properties have been measured or directly derived from measurements (Table 1 and Table S1). Miage Glacier is thus a "literature" site, the properties of which have been used by numerous other studies (e.g. Carenzo et al., 2016) and for a number of sites in this intercomparison (Table S1).

*6.1.2. Sites with poor performance*

In contrast to the European sites, **three sites stand out as low performance sites**: Djankuat, Lirung and Tasman. The intermediate and energy balance models cannot reproduce the observed melt at any of these three sites (Fig. 5) nor surface temperature at Lirung (Fig. S3). The only models that achieve a good or reasonable performance are the calibrated temperature index models (Fig. 5, Table 3), which tune their parameter(s) to maximise agreement with observed melt; it remains to be investigated whether their performance would remain high over a distinct validation period. The uncalibrated Hyper-fit model performed well, but it used literature parameters from the same site. It is not straightforward to disentangle the reasons leading to reduced model skills at these three sites. A low model performance can be associated with **either poor data** (input and/or validation data), **poor parameter values** (debris properties), or a **poor model**, (i.e. lacking or failing in the representation of processes that are important at those sites). At the three sites, the clustering of model performance shown in Fig. 6 suggests that either **all models miss a crucial process** that is important at those sites or there is a **common problem with the validation, forcing or debris properties data**, which affects equally all models except for the calibrated temperature-index models.

*6.1.3 Lirung study site and the difficulties in deriving thermal conductivity*

Lirung is one of the few sites with a conductivity value estimated from field data. The debris thermal conductivity value however is very high (Table S1). It was derived from thermistor records of temperature at variable depths within the debris using three approaches: i) the method by Nawako and Young (1982) and Brock et al. (2010), which assumes a linear vertical temperature gradient within the debris; ii) the approach by Conway and Rasmussen (2000) based on the diffusion equation; and iii) the approach of Anderson (1998), which assumes a sinusoidal variation of temperature in time and an exponential decay of temperature in space. All methods were applied to data collected in 2013 at an AWS location (unpublished data), and the value provided for this intercomparison experiment was the average of the three values. The values obtained with each approach differed considerably and were higher than many literature values, but since there was no way to establish which method was best, the average was provided. This points to an irony, that at one of the only sites where data were collected to estimate thermal conductivity, the values obtained through a devoted calculation may be inappropriate for modelling, pointing to a discrepancy between conductivity derived from field data and values needed by EB models, something that has also been suggested recently by Melo-Velasco et al. (2025). This suggests that: i) we **do not know yet which is the best method to**



**measure** or derive debris conductivity in the field, directly or from other field observations (Melo-Velasco et al., 2025), and
ii) simpler methods **providing bulk values**, such as the one by Brock et al. (2010), might be more suitable for the existing
energy balance models (e.g. the DEB model of Reid and Brock, 2010), which have been developed for conductivity values
derived in this way. Future efforts should therefore seek to devise methods to estimate debris thermal conductivity accurately
and in a manner that is consistent with their use in EB models.

It should also be noted that on Lirung there is **a difference of 14 cm between the debris thickness** at the location of the
automatic weather station (h = 30 cm) and the location of the ultrasonic depth gauge used as validation site (h = 44 cm). This
might explain some of the divergence of models' outputs from the observations. As observed in the Østrem curves for Lirung
(Fig. 10d), melt rates are lower (for all models) if thicker debris is used. We thus run one of the best performing energy balance
models, DEB$_{CF}$, with the debris thickness at the ultrasonic depth gauge location, with the debris conductivity and surface
roughness of Miage (k = 1.04 W m$^{-1}$ K$^{-1}$ and z$_0$=0.016 m, Brock et al. 2010), which were used by other sites as well (Table
S1), and with a combination of these changed debris properties. The model simulations with debris thickness changed from 30
640    cm to 44 cm differ by 22.2% and show a better agreement with the observations (Table 5). The difference from the standard
simulation is highest (42.7%) when we additionally consider the conductivity value from Miage (Table 5). This combination **reduces the total melt error of the DEB$_{CF}$**
**model at Lirung from 103.8% to only 16.6%,** demonstrating the large impact of inaccurate debris properties. The thicker
debris decreases melt rate considerably (and delays its peak), and the smaller conductivity value also considerably reduces and
delays the peak melt (Fig. S8). It is therefore **a combination of at least these two factors** (heterogeneous debris thickness
between the automatic weather station and validation site, and uncertainty in the site conductivity) that likely explains the poor
performance of all models at this site.

**Table 5.** Sensitivity test of melt rates by the DEB$_{CF}$ model at Lirung to substantially modified debris properties. Simulations with $h_d = 30$
cm (thickness at the automatic weather station location) and $h_d = 44$ cm (thickness at the ultrasonic depth gauge location), and with debris
conductivity and surface roughness from Lirung ($k = 1.55$ and $z_0 = 0.035$) and Miage ($k_d = 1.04$ and $z_0 = 0.016$), as well as combinations
of them. Debris properties underlined in *italics* denote the standard run (experiment 1) from this model intercomparison. Debris properties
in **bold** denote the largest difference with the standard simulations.

| Debris thickness, $h_d$ (cm) | *30* | | | | 44 | | | |
|---|---|---|---|---|---|---|---|---|
| Thermal conductivity, $k$ (W m$^{-2}$ K$^{-1}$) | *1.55* | | 1.04 | | 1.55 | | **1.04** | |
| Surface roughness, $z_0$ (m) | *0.035* | 0.016 | 0.035 | 0.016 | 0.035 | 0.016 | **0.035** | 0.016 |
| Total modelled melt (m w.e.) | *2.34* | 2.48 | 1.78 | 1.91 | 1.82 | 1.94 | **1.34** | 1.44 |
| Total modelled melt error (%) | *103.7* | 116.4 | 55 | 66.8 | 58.4 | 69 | **16.6** | 26 |
| Difference from standard simulation (%) | *0* | 6.0 | -23.9 | -18.4 | -22.2 | -17.1 | **-42.7** | -38.5 |
| Changed debris properties | - | $z_0$ | $k$ | $z_0, k$ | $h_d$ | $h_d, z_0$ | $h_d, k$ | $h_d, z_0, k$ |

*6.1.4 Tasman and Djankuat*

**At Tasman,** most models are clustered together (Fig. 6i), but a cold bias in surface temperature (with median RMSE 3.8°C,
Table S27) corresponds to a melt overestimation of about 58% (Table 3, Fig. 5). At this site, debris thermal conductivity is
very high (k = 1.8 W m$^{-1}$ K$^{-1}$) compared to literature values (Table S1). It was taken from Rohl (2008), who in turn used a
value from McSaveney et al. (1975) describing a pure mixture of rock and water without pore space. It is therefore inaccurate
for the conductivity of a porous debris layer, and likely responsible for the melt overestimation and cold bias. Even though
most models overestimate melt, an ablation stake at the same location measured higher total melt than that observed at the
ultrasonic depth gauge (Fig. S2i), suggesting the melt overestimation may be lower than reported. Finally, we cannot exclude



that at Tasman, which is the warmest and wettest of our sites (Fig. 2, Table S3), processes not included in the models, such as those related to the water content in the debris, might be playing a role.

Finally, **at Djankuat**, even the more empirical models in both their calibrated and uncalibrated versions fail to match the observed melt, with the exception of the calibrated KM1. Djankuat is the site with the second thickest debris cover (61 cm, Table 1) and the one most similar to the European sites in terms of conditions (temperature and relative humidity, Fig. 2). For this site, debris properties, and conductivity in particular, clearly play a key role in explaining the model failure. Debris conductivity is extremely high (2.8 W m$^{-1}$ K$^{-1}$), the highest of all sites (Table S1), and was taken from Bozhinskiy et al. (1986). Upon scrutiny, we realised that this is the conductivity of the rock material itself, and not that of a porous debris layer, which would be strongly reduced, reinforcing our conclusion that site-specific properties (representing the actual debris layer) are needed.

*6.1.5 Knowledge of debris properties remains a key gap*

An interesting site in comparison to all other sites is Changri Nup, where the overall performance is relatively high. At this site, the provided debris properties had been optimised to match observed melt with an energy-balance model not participating in the experiment (Table S1, Lejeune et al., 2013; Giese et al., 2020), thus explaining the high performance of most energy balance models (Fig. 5). The Changri Nup case exemplifies a relatively common strategy to **determine debris properties by optimization**, which seems a valid alternative when there are no reliable estimates from direct field measurements (Melo-Velasco et al., 2025). It seems particularly useful relative to literature values that may not be relevant (e.g. Tasman), or when direct methods provide divergent estimates with high uncertainty (e.g. Lirung). From all the cases considered in this intercomparison, and from the variety of approaches adopted to determine debris properties, it is apparent that **debris properties are not well constrained** at most sites, estimates from literature are often not appropriate, and even more importantly, that published methods to determine conductivity in the field (Conway and Rasmussen, 2000; Nicholson and Benn, 2012; Reid et al., 2012) **may not agree**, as exemplified by the case of Lirung, and confirmed by a separate study (Melo-Velasco et al., 2025). In addition, even if we are able to constrain debris properties at an individual automatic weather station, their values are affected by differences in porosity and pore water content across the debris areas of a glacier, and therefore are also likely to vary considerably in time and space. Neither aspect has seen much investigation to date. A lack of debris property data is also relevant for the GRO17$_B$ model, for which there are no data available of debris porosity and grain size, and which thus necessitated the calibration of the latent heat flux component. **Knowledge of debris properties** emerges thus from this intercomparison experiment as a **key knowledge gap** that the community should address, by both making a large community effort to compile and scrutinise existing estimates and datasets (e.g. Fontrodona-Bach et al., 2025), and by developing and thoroughly testing field methods to determine these crucial parameters.

## 6.2. Model performance, strengths and limitations

We have considered four groups of models ranging from empirical temperature-index models to physically-based energy balance models, with two models of intermediate complexity (Fig. 3). Our results indicate that **model performance varies greatly**, reflecting structural choices, distinct parameterisations of fluxes and levels of complexity in the representation of processes (Fig. 3 and Table 4). Energy-balance models offer a variety of model structures, flux calculations, temporal resolution, numerical solutions and vertical discretisation of the debris domain, and consequently produce a large range of model performance. Temperature index models perform well when calibrated, with a performance similar - and superior in some cases - to that of the energy balance models, and poorly when uncalibrated, although it was not possible to assess their performance over an independent evaluation period.





The discussion below is guided by the ranking of models in Table 4. We note however that an objective and unambiguous ranking is difficult to obtain, and this evaluation is valid for one melt season and the sites available to this intercomparison. Model choices respond to data requirements and specific research questions, and therefore our discussion does not disqualify models from being used for a particular research question or a given data availability.

**Energy-balance models**

*Large spread and variability among energy balance models' performance*

There are clear differences among the energy-balance models, with a large variability and spread, as well as a general difficulty to model melt at some sites. Models differ substantially in their calculations of both latent and sensible turbulent heat fluxes (Fig. 11 and 12). The two models with the highest degree of complexity according to our definition (see Sect. 4), $DEB_{CF}$ and
ROU15, were the highest ranked models based on their melt performance at the end of the modelling period (Table 4). However, the third most complex model, $GRO17_B$, clearly overestimates melt, especially at sites with thick debris. The $GRO17_B$ model, based on Evatt et al. (2015), includes the heat convection within the porous debris layer. Including this process adds complexity and uncertainty due to two extra parameters - the wind speed attenuation constant and friction velocity (Fig. 3) - which are required for the calculation of the evaporative (latent) heat flux. To obtain realistic estimates of these parameters,
they were calibrated against the bulk method of Nicholson and Benn (2006) for the calculation of turbulent heat fluxes (see Sect. S2.1, Table S6), but not to match melt observations as this was a requirement of this intercomparison experiment. The inclusion of this process is the main difference between $GRO17_B$ (including the latent heat flux within the debris) and $GRO17_A$ (not including the latent heat flux within the debris), and nevertheless showed no significant effect for experiment one of this intercomparison, with thickness prescribed based on actual measurements. This is likely because convection is most relevant
for thinner debris than at the sites considered in this experiment (Evatt et al., 2015). Its effect for thin debris is evident at Piramide, Tapado and Tasman for the model runs of experiment 3 (Fig. 10). The large melt overestimation at sites with thick debris is likely due to other model features, including the assumption of a linear temperature gradient in the debris layer. The same assumption is also made by the THRED and A-Melt models (Table S18), which are also ranked lower than the other energy balance models (Table 4), and have lower performances against surface temperature observations (Table S27, Fig. S6),
suggesting that the assumption of a linear temperature gradient may limit model accuracy, especially for thick debris and short time intervals (e.g. hourly).

*Disentangling differences between the best performing models*

Among the energy balance models, the models with the highest degree of complexity ($DEB_{CF}$, ROU15 and d2EB) show the best performance, excluding the problematic sites of Lirung, Tasman and Djankuat (see Sect. 6.2). These are also the models
with the highest consistency of performance against the two independent datasets. We thus further explore differences between these three best ranking models, as well as the $DEB_{PG}$, which is similar to $DEB_{CF}$ but has nevertheless a lower performance. The main differences between models are: i) the debris layer is discretized in a similar way in $DEB_{CF}$ and ROU15, but differently in d2EB and $DEB_{PG}$ (Table 2); ii) a snowmelt module is included in $DEB_{CF}$ and ROU15, but not in the other two models; iii) the rain heat flux is included in $DEB_{CF}$ and ROU15 but not in d2EB and $DEB_{PG}$; iv) the latent heat fluxes are
activated in an equivalent in manner in $DEB_{CF}$ and ROU15, which both assume debris is saturated when it rains, but differently in d2EB, which uses a parameterisation based on surface temperature (Steiner et al., 2018) and in $DEB_{PG}$, which assumes debris is saturated when the relative humidity of the air is 100% (Table S16); and finally v) stability corrections are included in $DEB_{CF}$ and $DEB_{PG}$, but not in ROU15 and d2EB (Table S15).

To evaluate the importance of each of these aspects, we explore the relative importance of these assumptions and flux
implementations using the $DEB_{CF}$ model, and compare the new simulations with the standard $DEB_{CF}$ run. The heat flux due





to rain is very small and neglecting it does not affect melt in any considerable manner at any of the sites (Table 6). Occasional snowmelt is present only at three sites, Arolla, Changri Nup and Pirámide, and important only at Changri Nup, where snow is present on the ground 8% of the time and snowmelt accounts for 47% of the ice- and snowmelt calculated with DEB$_{CF}$ (Fig. S13). In Arolla, snow and snowmelt are negligible, and in Pirámide snowmelt is very small (Fig. S13). The manner in which the debris layer is discretised does not seem to play a key role. Table 6 shows the effect of using two alternative approaches to discretise the debris: i) the approach by DEB$_{PG}$ and the one employed by Reid and Brock (2010). Both approaches lead to more numerous, thinner layers the thicker the debris, but neither approach has an effect on melt except for Tapado (5% difference), suggesting that the debris discretisation is slightly important only at thick debris (> 61 cm) sites.

**Table 6.** Results of changes in layer discretisation, LE calculations and assumption of precipitation flux obtained from running DEB$_{CF}$ and compared to the standard run. Values in italics in the header are the debris thickness in cm.

| % difference from DEB-model standard run | ARO | CN | DJA | LIR | MIA | PIR | SDF | TAP | TAS |
|---|---|---|---|---|---|---|---|---|---|
| *$h_d$ (cm)* | *6* | *10* | *61* | *30* | *22* | *18* | *11* | *80* | *30* |
| 1 cm layers | -0.02 | 0.00 | 0.78 | 0.52 | 0.37 | 0.03 | 0.03 | 5.35 | 0.41 |
| DEB$_{PG}$ layers (3 cm) | -0.11 | -0.07 | 0.28 | 0.01 | 0.07 | -0.01 | -0.33 | 4.84 | 0.27 |
| No precipitation flux | -0.06 | -0.18 | -0.04 | 0.23 | 0.00 | -0.02 | -0.01 | 0.00 | -0.03 |
| RH$_s$ = 100% when RH$_a$ = 100% | 1.10 | 10.14 | 21.64 | 1.49 | 2.79 | 10.79 | 1.14 | 0.00 | 1.87 |

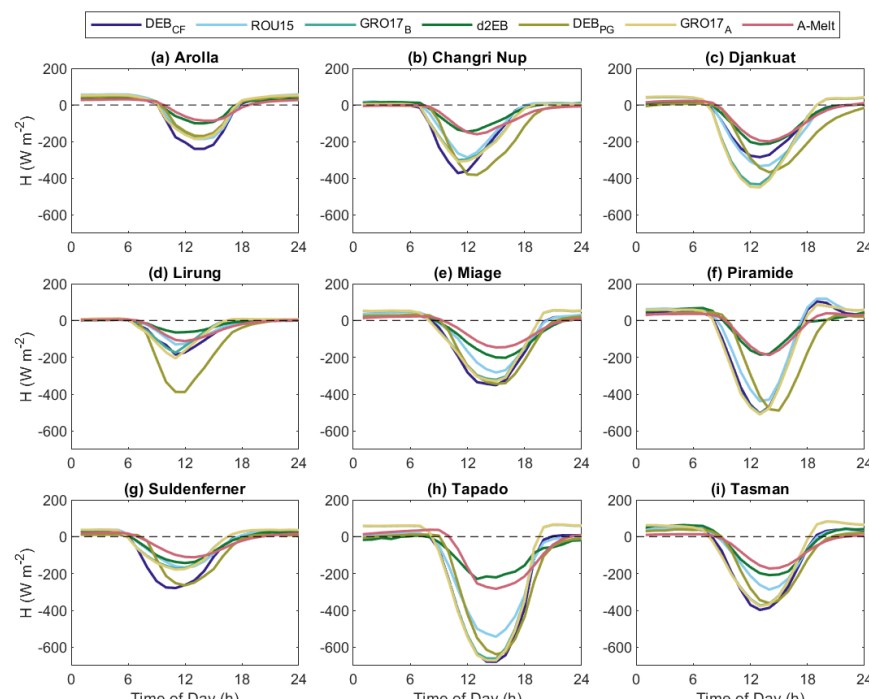

**Figure 11.** Average sub-daily cycle of the turbulent sensible heat energy flux (H) calculated by the energy balance models at all sites. Averages calculated over the entire simulation period are shown in Tables S28-S30.

The treatment of the turbulent fluxes seems instead a key element of distinction (Fig. 11 and 12). The use - or lack of - stability corrections can be a cause of large differences, but it was impossible in this experiment to quantify differences in total turbulent



fluxes or total melt attributable to the use of stability corrections as models differed in too many other assumptions to be able to isolate this effect. We notice here the large differences in the turbulent sensible heat flux (Fig. 11) and call for an experiment that can systematically identify their causes (see Section 6.3).

Key differences are evident also when considering **the latent heat flux**. The two models that use the same theoretical approach, $DEB_{CF}$ and $DEB_{PG}$, differ in the way the latent heat fluxes are activated and debris surface moisture is calculated. Since no observations of debris surface moisture were available, the two models used common (e.g. Rounce et al., 2015) but distinct assumptions. $DEB_{CF}$ assumed that water was available to evaporate at all timesteps during rain events (i.e. relative humidity of the debris surface is 100% during rain events), an assumption also made by ROU15. $DEB_{PG}$ instead assumed that there was water available to evaporate when the relative humidity of the air was 100%. As the latter situation barely occurred at our sites, the latent heat flux calculated by $DEB_{PG}$ is almost always zero (Fig. 12). The assumption about the relative humidity of the debris surface seems far more important than the inclusion of stability corrections (Table 6, Fig. 12). The ROU15 and $DEB_{CF}$ models, which differ in the manner the turbulent fluxes are calculated (Table S15 and S16) but use the same approach to determine the debris relative humidity (Table S16), produce similar turbulent fluxes at most sites, except for Djankuat (Fig. 12), which emerges from this intercomparison as a rather anomalous and problematic site (see Sect. 6.1).The d2EB model includes an empirical parameterisation to determine the relative humidity of the surface from the vapour pressure of the air (Steiner et al., 2018, and Sect. S2.1), which requires two empirical parameters originally estimated from 10 days of measurements on Lirung glacier in autumn, and used at all other sites. The latent heat flux simulations with this model are distinct from all others, and from $DEB_{CF}$ and ROU15 in particular (Fig. 12).

We have no way to evaluate which of those formulations is the most correct besides doing so indirectly through assessment of melt and surface temperature accuracy. The fact that $DEB_{CF}$ has one of the highest performances, and that some of the other characteristics of this model seem to have little influence (Table 6), seems to suggest that the approach to calculate the turbulent fluxes adopted in that model might be the most suitable for the whole range of sites based on the measured and assumed debris properties. Neglecting latent heat fluxes will likely lead to overestimation of the sensible heat fluxes (Collier et al., 2014; Nicholson and Stiperski, 2020), so it seems crucial to validate the flux calculations directly, and not indirectly as in this intercomparison experiment (see Sect. 6.3). Measurements of latent heat fluxes are becoming available (Steiner et al., 2018; Nicholson and Stiperski 2020), creating the opportunity for such an effort.



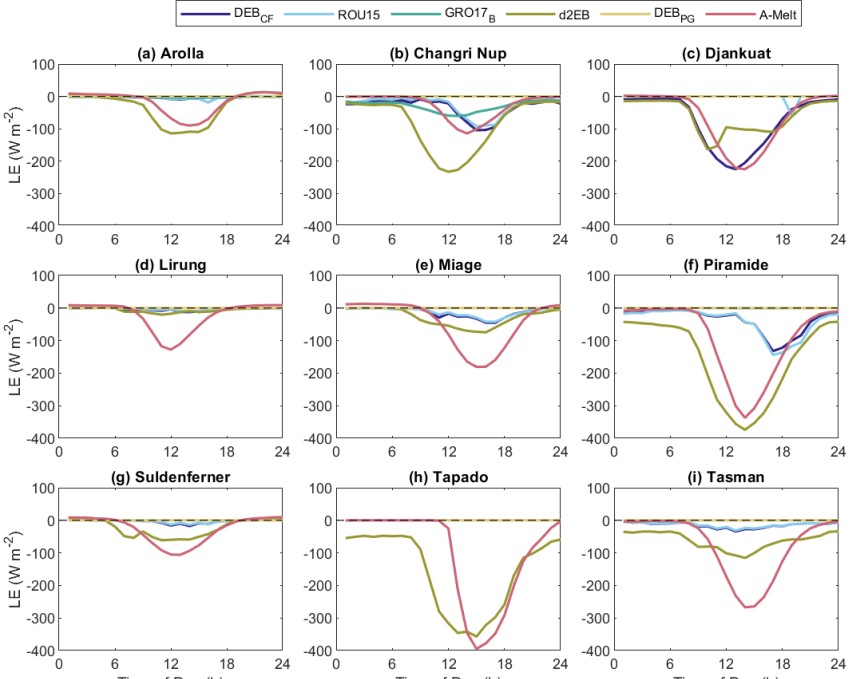

**Figure 12. Average sub-daily cycle of the turbulent latent energy flux (LE) calculated by the energy balance models at all sites. Averages calculated over the entire simulation period are shown in Table S28-S30. Lines are not visible when the flux is close to zero and models overlap each other.**

**Temperature index models**

Temperature index models offer computational efficiency. In this experiment, they ran on average $10^3$-$10^4$ times faster than energy balance models (Table S26), a substantial advantage when used for spatially-distributed or long-term simulations. In addition, they require only a few, commonly available input data. We show that they perform well in reproducing daily melt rates when calibrated against melt observations from the same time period, but also note that no independent validation was possible. The Hyper-fit model shows that empirical models can perform reasonably well with parameters determined from independent sites and time periods on the same glacier. Since they rely on air temperature as input, they cannot reproduce sub-daily sub-debris melt rates accurately and the strong sub-daily variability that characterises melt rates under thin to moderate debris (Fig. S9). This is something that has been suggested already for debris-covered ice melt (Carenzo et al., 2016) and is well known for debris-free ice melt (e.g. Gabbi et al., 2014).

Our results show that calibration greatly improves the performance of temperature index models. The simpler the model, the more important its calibration, as most of the variability in melt rates and the processes controlling them is represented by the few parameters. The KM1 model performs considerably better than its uncalibrated version KP1 (Fig. 7, Table 3). The Hyper-fit model has a stronger performance when calibrated than uncalibrated (Fig. 7, Table 3). The uncalibrated version uses parameters derived for the same glacier as in this intercomparison but for time periods distinct from those of the intercomparison. This differs from the other uncalibrated models which use a single parameter set applied to each study site. Nonetheless, the uncalibrated version of the Hyper-fit model reveals that parameters determined outside of the study period and from other locations on the same glacier can produce consistent high model skill (Fig. 7, Table 3), at least within the experimental setup of this model intercomparison. This means that for glaciers where Hyper-fit model parameters ($DDF_{ice}$ and h*) have been constrained in the past, sub-debris melt may be modelled reasonably well (Table 3). This ability of the Hyper-



fit model is possible because of the stability of the characteristic debris thickness, h* in space and time across individual glaciers.

The calibration strategy chosen by each modeller has a strong impact on model performance. How empirical models were calibrated was left to modellers as part of the experimental design. These models were calibrated both against the cumulative daily melt, sub-period melt and total melt, the latter being the metric we use for their evaluation (see Sect. 4.3). It is clear that if the error metric chosen to evaluate model output is different from what the model was tuned to then the performance would be different, and possibly reduced. Use of a different calibration metric might result also in distinct parameter values, something to remember when considering parameter transferability. As an example, we recalibrated the KM1 model based on daily melt, rather than total melt, and this resulted in lower performance against total melt at many sites (Fig. S12), but an increased ability to reproduce the variability during the period of simulations (e.g. Changri Nup, Lirung), suggesting that at some sites relatively accurate calculation of total melt might result from compensation of errors in the daily melt rates.

**Models of intermediate complexity**

The two models of intermediate complexity, SEB and ETI, perform similarly to that of the best energy balance models (Table 4 and Fig. 7). Both models calculate net shortwave radiation flux using identical inputs, with the DETI having three empirical parameters and the SEB using two empirical parameters and debris conductivity to calculate heat conduction through the debris (Fig. 3). A strength of both models is their ability to reproduce the sub-daily cycle of melt rates, similarly to energy balance models (Fig. S9), an ability that derives from inclusion of the shortwave radiation balance. Crucially, both models require calibration against variables that exhibit sub-daily variability. However, the uncalibrated $DETI_m$, using parameters from Carenzo et al. (2016) for Miage glacier, performs comparably to its calibrated version across sites (Fig. 7, Table 3). This suggests suggests that the parameter values derived from Carenzo et al. (2016) are fairly transferable across sites and that calibration may not significantly enhance performance if the energy balance model to calibrate against is suboptimal.

### 6.3. Future perspectives

Cooperation between a large number of scientists has allowed this intercomparison to provide a robust assessment of different approaches to modelling sub-debris melt at glaciers across the world. It has also highlighted model and data limitations and provided insight into what data should be collected to enable a future model intercomparison focused on the physical processes that control melt in different climatic settings.

*A new vision for old measurements*

While it is clear that the growing debris-covered glaciers community is making a large effort to collect data at a variety of sites (Fontrodona-Bach et al., 2025), it is still young in terms of data collection and standardisation. Meteorological data quality has been a major issue as no data provider was able to establish the measurement accuracy beyond the sensor accuracy. It seems important to design observations where both the accuracy of the measurements and the actual meaning of a measurement are clearly defined. Despite recent advances in deriving debris properties (Rounce et al., 2015; Quincey et al., 2017; Miles et al., 2017, McCarthy et al., 2017; Nicholson and Mertes, 2017; Nicholson et al., 2018,  Bisset et al., 2022; Messmer and Groos, 2024), debris properties remain a very major gap in knowledge, as discussed in Sect. 6.1. Comparable, standardised field and laboratory measurements should be designed to determine debris properties, and debris thermal conductivity in particular, and their variability in space, time and with depth, under distinct meteorological and surface conditions.

A discrepancy that should be considered is that often the three sets of data needed for model simulations (input meteorological data, debris properties and validation data) are measured at different locations. Considering the high heterogeneity of debris thickness and properties even within short distances (e.g. Nicholson and Benn, 2012), it is important to account for these spatial



discrepancies. For instance, the point where automatic weather station sensors 'look at' is often metres distant from the ablation stake or ultrasonic depth gauge measuring melt. Melt records are often measured at only one site because of the effort required and can be non-representative of the area around automatic weather stations. This has been exemplified by the case of Lirung, where differences in debris thickness between the location of the AWS and ultrasonic depth gauge affected the comparison of

modelled and observed melt rates.

*Recommendations for novel measurements*

Despite the strong need for standardisation and improvements in the quality of existing measurements, more routine measurements and estimates of their uncertainty, it seems also crucial that new datasets are collected that allow both internal validation of existing models and model developments. Measurements of internal variables allowing for the testing of state

variables and intermediate fluxes simulated by energy balance models (e.g. debris temperature profiles, water content, single fluxes estimates) should be collected. What has been called "internal validation" is since a decade or longer widely recognised and accepted in hydrological research, and increasingly also in glaciological mass balance and runoff models (Huss et al. 2008, Ragettli et al., 2015). It should be adopted widely also by the debris-covered glaciers research community.

It seems also crucial to collect accurate datasets at sub-daily resolution to evaluate the performance of all models, and energy

balance models in particular, that can work at high temporal resolution, as part of this strategy of internal validation and to avoid equifinality problems (Gabbi et al., 2014). Ultrasonic depth gauge (UDG) records do in theory offer sub-daily records, but these are noisy and there is little agreement as to how those datasets should be treated and processed, nor about what is their actual accuracy at hourly or higher resolution time intervals. Supplementary information (e.g photos of the station/UDG, measurements from the surface to UDG at start and end of the data set) are needed to confirm the accuracy of UDG data. A

systematic collection of additional independent validation datasets such as within-debris temperature and measurements of debris moisture seem important (Sakai et al., 2004) to allow validation of internal processes and thus testing of individual model components, and not only bulk simulations. Community compilation of manuals of best practice guidance for observations may be a useful way forward to increase the systematic collection of relevant data.

We have shown that there are still major knowledge gaps, and models are in need of further testing and developments, even in

a field relatively well established such as melt modelling at the point scale. At this scale, the uncertainties associated with the spatial variability of meteorological data and debris properties are removed, and still uncertainties are large. It seems particularly important to design acquisition of observations to test the model component proposed by Evatt et al. (2015) to account for air flow and heat convection into a porous debris layer and to allow evaluation of the schemes designed to estimate the turbulent heat fluxes (with e.g. eddy-covariance measurements as in Collier et al., 2014). Similarly, water flow through the

debris matrix should be accounted for (Fyffe et al. 2020), as a saturated layer thickens and thins with daily melt cycles and in response to warm and cold weather, causing temporal variations in both debris conductivity and the evaporative flux.

A key aspect of investigation that is left open is a thorough understanding of the relative importance of processes. This should be systematically assessed for both distinct debris thicknesses (and thus debris properties) and climate conditions. We were able to show that the conditions for latent heat flux activation are more important at some sites (Changri Nup, Djanukuat and

Pirámide, with errors of 10 to 25% in total melt, Table 6) than at others. Those three sites however are remarkably distinct, and include: i) a high elevation, cold, humid (80% average relative humidity) site with relatively thin debris (10 cm), Changri Nup; ii) a relatively warm site with average relative humidity of 60% and thick debris (61 cm); and iii) a relatively cold, very dry (30% average relative humidity) Andean site with 18 cm debris. It is thus almost impossible to identify which characteristics make these three sites sensitive to changes in the parameterisation of debris surface moisture content. A devoted

field experiment, where each of those factors (debris thickness, debris moisture and water flow, climatic controls) are controlled, should be designed to disentangle the driving factors.





An aspect of model performance that has emerged from this intercomparison as worth further investigation is the dependence of model skills as a function of debris thickness. With increasing debris thickness, processes such as heat storage and conductivity might become more relevant, while others such as the air flow within the debris and associated convective heat

exchange with the underlying ice (Evatt et al., 2015) might lose importance. Both thick and thin debris should be investigated at sites if possible. It is also clear from our analysis that at some sites, and Lirung in particular, thin debris is where models diverge most, and we are not able to constrain an Østrem curve with a model ensemble. Areas of thin debris, usually located in the upper region of the debris-covered area, is where maximum rates of sub-debris ablation are predicted (Fyffe et al., 2020) and where determining ablation rates with confidence is critical to pin down the interplay between melt and ice flux in

determining the glacier profile (e.g. Nicholson and Benn, 2012). This consideration should guide data acquisition to explain the model divergence for thin debris across sites.

*Suggestions for future model developments*

The models that joined this intercomparison experiment reflect the state-of-the-art of modelling skills and structures available to the community. There is still large potential for model improvements, however. No model in this intercomparison includes

freezing/refreezing and the cold content of the frozen porous debris layer. Although this may not be relevant during the ablation season, it may be important at high elevation sites and during shoulder seasons (Giese et al., 2020). On Lirung Glacier, for instance, Steiner et al. (2021) demonstrated that during up to a week in spring all energy available was used to defrost the debris. A second crucial limitation of all models is that none of them considers water content in the debris, which has been shown to impact modelled melt volumes (Collier et al. 2014). A key aspect related to this is the lack of observations of debris

water content, either for model development or validation. Refreezing and water content in the debris should therefore be included in future model developments (e.g. Winter-Billington et al. 2022).

Snow has been neglected from this intercomparison, which as a first step and to allow the inclusion of the largest possible number of models, has focused on the ablation season. For calculation of mass balance and long-term glacier changes, however, it is important to run models for an entire year and then for multiple years. This would require inclusion of snow accumulation

and melt above the debris layer.

Finally, this intercomparison has evidenced the need to assess how important the heat convection into a dry porous layer is (Wicky and Hauck, 2020), when it occurs, and at which sites it is dominant. This would allow evaluating the relative importance of the model developments suggested by Evatt et al. (2015).

*Limitations and recommendations for Phase II*

There are a number of limitations to our study, which could be addressed in a phase II of this experiment. First, we have evaluated model skills against **low temporal resolution melt observations**, daily at best, which cannot reproduce the diurnal cycle of melt, and from sensors that might have considerable errors. The results are striking, as the highest ranked models are calibrated temperature-index models (Hyper-fit, KM1 and DDF$_{debris}$, Table 4) that cannot reproduce the daily cycle because of intrinsic structural limitations (Fig. S9). While at the seasonal scale the sub-daily variability may not matter, model

intercomparisons should ideally be carried out **at a variety of scales**, which each correspond to a distinct model ability and model purpose (e.g. mass balance versus runoff simulations). It also points to the fact that we do not have in most cases measurements of high resolution available, and improved melt measurements at daily resolution (rather than weekly/seasonal stake measurements) would be beneficial.

Second, this study has made a large effort to gather existing data sets across the globe so as to include in the comparison as

many sites with distinct debris and climate characteristics as possible. However, all our conclusions rely on data **from only one melt season** at each of the nine sites. Although some uncalibrated temperature-index models performed well using





literature parameters, this limitation did not allow testing the model robustness and parameter transferability in time over multiple melt seasons. To fully assess and equally compare the performance and transferability of empirical temperature-index models and energy balance models, and their robustness in space and time in particular, we suggest that a follow-up effort should test the model performance of calibrated models over an independent validation period. In general, model simulations should be tested over multiple years to assess their robustness and how the errors we identified over one melt season propagate (or not) over the temporal scales needed for mass balance change calculations.

Third, it was apparent from this intercomparison that the broad variety of choices in model structures, coding languages, debris discretisation, parameterisations, temporal resolutions and assumptions masked at times actual differences in representation of processes. This hindered an unambiguous identification of strengths and limitations, as too many differences in model choices did not allow to identify the actual reasons for distinct model performances. A lesson learned from our experiment is thus that a consistent modelling framework including multiple parameterisations and modelling assumptions is needed to identify the importance of individual processes and assumptions. We would thus advocate for such a modelling framework that includes the suites of models tested in this experiment, as well as the processes we identified as missing above.

## 7. Conclusions

We have compared 14 models of different complexity to simulate glacier ice melt under debris (Fig. 3), including energy balance models, temperature index models and models of intermediate complexity. The models were run at 9 sites across the globe, for one melt season of variable duration, and validated against sub-debris ice melt and surface temperature observations. Our main conclusions are as follows:

- Energy balance models and empirical temperature index models perform in a distinct manner and serve distinct purposes. In general, temperature index models perform very well (median performance) when calibrated, and poorly when applied in their uncalibrated form. However, the Hyper-fit model and the DETI model show that site-specific literature parameters determined outside of the study period can produce viable melt estimates.

- Energy balance models show a range of performance and model skills. A clear finding from this work is that the models that perform best are those with the highest degree of complexity at the debris-atmosphere interface. We were not able to demonstrate the added value of additional complexity within the debris, because of lack of data representative of processes within the debris layer. The use of simplifying assumptions (and of a linear temperature profile within the debris in particular) within the model that included convection in the debris made it difficult to disentangle the importance of this process.

- A striking result of our intercomparison is that models perform well at the three Alpine sites of Arolla, Miage and Suldenferner, and consistently poorly at Djankuat, Lirung and Tasman. While a variety of reasons contribute to explain this inter-site variability in model skill, we have been able to clearly identify variable knowledge of debris properties, and thermal conductivity in particular, as a key cause.

- Debris properties are a major control on melt simulations and model performance. We thus need estimates of debris properties (especially surface roughness and thermal conductivity) of high accuracy. Crucially, we currently are not able to constrain them using measured values at most sites. The uncertainty ranges of debris properties typically used in the literature are insufficient for most models to encompass observed melt values. Efforts should be devoted to measure debris properties in the field and establish their variability in time, space and with depth.



- Despite tremendous advances in recent years, we showed that sub-debris melt models still need to be improved, and important model developments are needed. Most are not able to cope with debris-snow interactions, moisture in the debris is not accounted for and refreezing rarely represented.

We suggest that a follow-up to this intercomparison should focus on systematically improving a single model framework (written in the same language, with the same assumptions and main schemes) to understand the effects of individual processes such as debris moisture related processes (phase change, percolation, conduction properties change) and convection within the debris. This would allow to both: i) determine the most appropriate representations and parameterisations of each process; and ii) understand where the balance lies between increasing the number of parameters and achieving substantial improvements in model skills, thus also avoiding equifinality problems and error compensations that might be related to a too high number of parameters. Adding more complex process representation bears the risk of transferring uncertainty into parameters that might be impossible to measure. It also seems imperative to establish a hierarchy of processes in terms of their importance, and to focus on advanced, robust representations of those. A way forward is thus to develop a flexible, modular model that would allow testing distinct calculations and parameterisations within the same model structure, allowing comparisons of single model components.

Our main conclusions are that models need to be improved and knowledge of debris properties substantially advanced. Data collection is often regarded as functional more to model validation than development. We suggest that data collection and model development should be closely coupled, iteratively informing each other - something that only a truly community, cooperative effort might afford. In addition to the model development needs identified above, we need to clearly identify community priorities and methodologies for making debris measurements.



*Data Availability.* All data were stored on the Zenodo community on debris-covered glaciers (zenodo.org/communities/iacswgondcgs) set up by the working group on debris-covered glaciers from the International Association of Cryospheric Sciences. The data are publicly accessible at:

- https://zenodo.org/records/3047649 (Arolla)
- https://zenodo.org/records/3048780 (Changri Nup)
- https://zenodo.org/records/3049871 (Djankuat) (Rets et al. 2019)
- https://zenodo.org/records/3050327 (Lirung)
- https://zenodo.org/records/3050557 (Miage)
- https://zenodo.org/records/3056072 (Piramide)
- https://zenodo.org/records/3056524 (Suldenferner)
- https://zenodo.org/records/3362402 (Tapado)
- https://zenodo.org/records/3354105 (Tasman)

Model simulations were submitted individually and are also available on Zenodo at https://doi.org/10.5281/zenodo.15754455.

*Author contributions.* FP, AFB, DR, CF are the core authors. The rest of the authors are in alphabetical order. FP and DR had the initial project idea, developed the experimental set up and recruited modellers and data providers. AFB and FP coordinated the project. FP wrote most of the manuscript and gave constant input to the data analysis and manuscript preparation. AFB managed the experiment data, did most data analyses, and contributed to writing the manuscript. CF contributed to the data analysis. LA, AFB, KF, CF, PG, ARG, WI, CM, MM, ER, DR, AS, JS and AWB provided model simulations. AA, BB, WI, SF, SM, JM, EM, HP, TS, JS, and PW provided observational data. AFB, FP and ER prepared and processed observational data. All authors commented on the manuscript.

*Competing interests.* The authors declare that they have no conflict of interest.

*Acknowledgments.* This project has received funding from the European Research Council (ERC) under the European Union's Horizon 2020 research and innovation programme grant agreement No 772751, RAVEN, "Rapid mass losses of debris covered glaciers in High Mountain Asia". It was also supported by the SNSF RENOIR project "Resolving the thickness of debris on Earth's glaciers and its rate of change (RENOIR)", project number 204322.

David Rounce received support from NASA-ROSES program grants NNX17AB27G and 80NSSC17K0566.

Walter Immerzeel and Jakob Steiner acknowledge support from the European Research Council (ERC) under the European Union's Horizon 2020 research and innovation program (grant agreement no. 676819).

Ben Brock acknowledges support from the EU/FP7 ACQWA (Assessing Climate impacts on the Quantity and quality of WAter) project, NERC grant NE/C514282/1, the British Council-Italian Ministry of University and Research Partnership programme and the Carnegie Trust for the Universities of Scotland.

The authors acknowledge the International Association of Cryospheric Sciences (IACS) for supporting the creation of the Debris-Covered Glaciers Working Group (DCG-WG) which enabled this model intercomparison experiment.



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
