# Peer review of "DCG-MIP: The Debris-Covered Glacier melt Model Intercomparison exPeriment"

_EGUsphere, 2025_

## Author Comment (AC1)

Black: reviewer's comments (RC1.1 to RC1.15)

Blue: authors' response

In the case of proposed revised text: "Normal blue for original text, red strikethrough for removed text and bold purple for revised text"

RC1: 'Comment on egusphere-2025-3837', Anonymous Referee #1, 18 Sep 2025 RC1.1 This is an eagerly awaited report of results from the debris-covered glacier intercomparison project. It has a good geographical range and number of participating models, but is limited to only one melt season per site. This does not allow spin up of debris and ice temperatures or division into calibration and validation periods. Repeating the experiments with more years of data would be beyond feasible modifications for this paper, but reasons for and implications of this restriction should be discussed in more detail.

We thank the referee very much for their review of our manuscript and thoughtful comments, which will substantially improve the paper. We provide below a detailed action plan for a revised manuscript.

We agree with the reviewer that a limitation of our study is that the model comparison was carried out over only one melt season at each site. This was, as the reviewer notes, dictated by data availability, as very few sites had measurements available for more than one season. We thus appreciate that the reviewer finds it unfeasible to repeat our experiments for more years at this stage, which would indeed be impossible. Instead, we have, as suggested, expanded the discussion of this limitation in the manuscript. In the original submitted paper we had devoted a sub-section of the discussion (Sub-section 6.3, *Limitations and recommendations for Phase II*) to the experiment's limitations, where we included a paragraph on this and a detailed recommendation for a follow-up experiment addressing this issue (Lines 919-927). In addition to that section, we propose to include a description of this issue in the model evaluation section:

Addition at the end of line 404 (Model evaluation): "Model performance was only evaluated over one ablation season because data were not available for additional seasons at most sites. This limited our capacity to assess model robustness over multiple seasons. For the models requiring calibration, i.e. temperature-index, enhanced temperature-index and simplified energy-balance models (MCC19, DETI, Hyper-fit, KM1, DDFdebris), in particular, this does not allow a separate validation period, as all models requiring calibration were optimised for the entire period of data available."

We will also indicate in Section 2. Experimental setup that energy balance models were not spun up prior to the simulations for the experiment, given the lack of data:

Addition at the end of Line 170 of the submitted manuscript: "Given that data were available for one melt season only, no spin-up was possible for energy balance models. Given that no snow was present at the beginning of our experiment, that the time required to spin up within-debris temperature should be in the order of hours or

a few days, and that only one model allows ice-temperatures to go below zero, we expect the lack of a spin up period to have a minimal effect on the study."

**RC1.2** For annual mass balance simulations, temperature-index models also require precipitation as input.

It is true that temperature-index models also require precipitation as input when used to simulate glacier mass balance. However, in the paper we focus only on melt processes and the ability of models to simulate melt under debris during the melt season only. For the simulations carried out in the paper, precipitation is therefore not required.

We will specify that this sentence only regards melt modelling as follows:

"Despite this, temperature-index models have seen successful applications at the glacier and regional scale because they are simple, computationally efficient and require only air temperature (occasionally incoming shortwave radiation) as input **to model melt** and a low number of parameters (e.g., Kraaijenbrink et al., 2017)."

**RC1.3** Table 1: Position to four significant figures locates the glaciers to within about 10 m.

The coordinates refer to the automatic weather station, and not a general glacier location. Proposed revised caption:

"Table 1. Overview of study sites. Validation data indicates what kind of melt observations are used at each site: ultrasonic depth gauge (UDG), ablation stakes, a draw-wire, and debris surface temperature (Ts). hd = debris thickness. The latitude and longitude coordinates refer to the locations of the automatic weather stations."

**RC1.4** Table 2: With net solar radiation used as an input, why is KO2 not classified as an enhanced temperature index model?

This is a very valid point and the reviewer is right that KO2 should be considered as an enhanced temperature index model, despite not resolving the daily cycle of shortwave radiation because of its daily time scale. We will modify the manuscript as suggested, applying the appropriate changes to figures and tables. We retain however our rank in terms of model complexity. According to our definition of complexity (Figure 3), DETIm and KO2 are both enhanced temperature models, both using solar radiation as input, but as DETIm is run at hourly resolution, we regarded DETIm as more complex than KO2.

**RC1.5** Figure 4: Notation for radiation fluxes differs from Equation 1.

Thank you for spotting it. We will standardise these notations.

**RC1.6** 272: Elsewhere it is stated that net shortwave radiation is given, not calculated.

It is indeed given, not calculated. We will change the text as follow:

"All energy balance models directly use the provided observed calculate the net shortwave radiation flux, and calculate the longwave radiative fluxes and the turbulent sensible heat flux at the surface (Fig. 3).

**RC1.7** 282: Relative humidity is a property of air. Are the assumptions rather on the wetness of the debris surface?

Yes, the reviewer is right: the assumption is on the wetness of the debris surface, based on the relative humidity of the air. We propose the following revised text to clarify this:

"Since no data on the water content within the debris were available at any of the sites, modellers either neglected this flux or made assumptions on the actual relative humidity of the air at the debris surface (RHs) based on the relative humidity of the air or precipitation occurrence (Table S16). These vary from assuming that the air at the surface is saturated when it rains (DEBCF, ROU15) to assuming that RHs=100% when the air relative humidity at the measurement height (RHa) is 100% (DEBPG)."

**RC1.8** Figure 5: (a) and (b) labels are missing from the figure.

Thank you, we will add these.

**RC1.9** 458-474: With this much discussion of Figure S3, it would be better to include it in the paper.

Thank you for the suggestion. We agree that Figure S3 is extensively discussed and could be included in the paper. We will include it together with Figure 8 arranged as two panels, as suggested below:

Figure R1.1. New revised Figure 8 including Figure S3 as bottom panel.

RC1.10 577: Djankuat is just Fig. 10c.

We will correct this.

**RC1.11** Figure 11: A reminder that fluxes are negative when away from the surface would be useful in the figure caption.

We will add this in the caption.

**RC1.12** 795: The "uncalibrated" version of Hyper-fit with parameters for the same glacier but a different time period would be regarded as a calibrated model in any other study.

We agree that the term "uncalibrated" is ambiguous here and might be misleading. Since we left the definition of uncalibrated up to the modellers, the Hyper-fit uncalibrated parameters were estimated by the modeller using Ostrem curves derived from debris thickness and melt data from the same glacier (except for two glaciers, Suldenferner and Tasman for which global data were used), but without using the specific dataset from this experiment. To avoid confusion, we propose to replace the "Hyper-fit uncalibrated"

version with "Hyper-fit estimated" version. We will correct this in the description of this model in the supplement, and provide the definition of estimated in the main text.

**RC1.13** I have not checked the reference list in detail, but the authors should. Kuzmin (1961) at least is missing.

We will check and revise the reference list.

**RC1.14** Table S1: Although the models are not required to calculate albedo, it would be interesting to know the measured debris albedo for each site.

We will add the mean measured albedo at each site.

**RC1.15** The paper is well written, with few errors that I noticed: 73: "which has has"; 821: "This suggests suggests"

Thank you for spotting these, we will correct them.